# Unbiased Recommender Learning from Implicit Feedback via Weakly Supervised Learning

**Hao Wang** [1]  **Zhichao Chen** [1]  **Haotian Wang** [2]  **Yanchao Tan** [1]  **Licheng Pan** [1]  **Tianqiao Liu** [3]  **Xu Chen** [4]
**Haoxuan Li** [✉ 5]  **Zhouchen Lin** [✉ 6 7 8]

## Abstract

Implicit feedback recommendation is challenged by the missing negative feedback essential for effective model training. Existing methods often resort to negative sampling, a technique that assumes unlabeled interactions as negative samples. This assumption risks misclassifying potential positive samples within the unlabeled data, thereby undermining model performance. To address this issue, we introduce WeaklyRec, a model-agnostic framework that reframes implicit feedback recommendation as a weakly supervised learning task, eliminating the need for negative samples. However, its unbiasedness hinges on the accurate estimation of the class prior. To address this challenge, we propose Progressive Proximal Transport (PPT), which estimates the class prior by minimizing the proximal transport cost between positive and unlabeled samples. Experiments on three real-world datasets validate the efficacy of WeaklyRec in terms of improved recommendation quality. Code is available at https://github.com/HowardZJU/weakrec.

## 1. Introduction

Recommendation systems aim to capture user preferences through their feedback, subsequently offering personalized content or product suggestions. They find extensive applications across various domains, including e-commerce (Smith and Linden, 2017; Wang et al., 2022), advertising (Zhou et al., 2018), and entertainment (Gomez-Uribe and Hunt, 2016). By accurately predicting user preferences, recommendation systems play a pivotal role in enhancing user experience and increasing business revenues (Gao et al., 2022a;b; Zhang et al., 2025; Zheng et al., 2025).

The core of recommendation system is the handling of user feedback, which can be broadly categorized into explicit and implicit types. Explicit feedback, such as ratings, comments, or like/dislike tags, reflects user preferences faithfully but is challenging to collect, as it requires active user participation, potentially diminishing user experience. Conversely, implicit feedback, such as page views or click logs, is readily available and does not require active user participation. Therefore, recommendation with implicit feedback, termed as ImplicitRec, has gained significant attention in industrial applications (Togashi et al., 2021; Lee et al., 2022).

A unique challenge in ImplicitRec is missing negative feedback (MNF) (Lim et al., 2015; Yang et al., 2018). Specifically, the available data only consists of positive and unlabeled samples. This absence of negative feedback complicates the task of differentiating between items that are either disliked or simply unexposed to the user across the unlabeled interactions (Zhou et al., 2021; Ren et al., 2023; Wang et al., 2021). For instance, a user's failure to click on a video might not necessarily indicate disinterest; it could simply mean the user was not exposed to that particular video. Treating unlabeled interactions as negative instances (Zhou et al., 2022; Kang and McAuley, 2018) ignores the existence of positive samples within the unlabeled data, thereby introducing bias and compromising recommendation performance (Wang et al., 2021).

To counteract the MNF issue, one line of works advocate downweighting unobserved samples (Hu et al., 2008; Liang et al., 2016), employing pairwise ranking loss (Rendle et al., 2009). While promising, these methods are largely heuristic and lack a solid theoretical foundation for unbiasedness. Recent works have shifted focus towards the construction of theoretically unbiased estimators of the ideal risk, using only positive and unlabeled samples. Representative methods such as RMF (Saito et al., 2020), UBPR (Saito, 2020) and CJMF (Zhu et al., 2020) primarily leverage propensity re-

---

[1]Zhejiang University, [2]National University of Defense Technology [3]TAL Education Group [4]Renmin University of China [5]Center for Data Science, Peking University [6]State Key Lab of General AI, School of Intelligence Science and Technology, Peking University [7]Institute for Artificial Intelligence, Peking University [8]Pazhou Laboratory (Huangpu), Guangzhou, China. Correspondence to: Haoxuan Li <hxli@stu.pku.edu.cn>, Zhouchen Lin <zlin@pku.edu.cn>.

*Proceedings of the 42nd International Conference on Machine Learning*, Vancouver, Canada. PMLR 267, 2025. Copyright 2025 by the author(s).

weighting, a methodology rooted in causal inference (Li et al., 2023b), which offers unbiased estimates provided accurate propensity score estimation (Li et al., 2023c;a). However, identifying accurate propensity scores remains an elusive goal in ImplicitRec[1]. Furthermore, the derived risk estimators do not necessarily maintain non-negativity (Zhu et al., 2020; Saito, 2020; Saito et al., 2020), posing the risk of non-convergence. Consequently, the MNF issue remains an open challenge in ImplicitRec that warrants investigation.

This work introduces Weakly Supervised Recommendation (WeaklyRec), a model-agnostic framework based on learning from weak supervision paradigm (Sugiyama et al., 2022), to develop rigorous formulations and practical solutions for handling MNF without reliance on vulnerable propensity scores. Specifically, we first reformulate the ideal risk and construct an empirical estimator that obviates the need for negative feedback. We show that this estimator is unbiased, consistent with the ideal risk, and amenable to optimization via stochastic gradient descent, with unique attainable optimum for representative recommender families. However, the estimator's performance is contingent upon a predefined class prior, which requires expert knowledge for empirical selection. To address this, we develop a Progressive Proximal Transport (PPT) method which achieves automatic class prior estimation by minimizing the proximal transport cost between the positive and unlabeled samples. Extensive evaluations on three publicly available real-world datasets substantiate the efficacy of the proposed method.

The main contributions of this paper are summarized below:

- We propose WeaklyRec, a model-agnostic framework that addresses the ImplicitRec problem using weakly supervised learning. Its core component is a risk estimator that eliminates the need for negative samples and is theoretically unbiased with respect to the ideal risk, provided that accurate class prior estimation is available.

- We introduce PPT, an optimal transport-based strategy for accurate class prior estimation, which is a necessary factor to ensure the unbiasedness of WeaklyRec.

- We validate the efficacy of WeaklyRec on three real-world datasets, where the recommendation performance is improved and the class prior is accurately estimated.

## 2. Preliminary

In this section, we establish technical preliminaries essential for comprehending the proposed methods. Initially, we delineate the fundamental symbols and formalize the problem

definition of ImplicitRec. Subsequently, we offer a concise overview of optimal transport, a mathematical framework that is intrinsically linked to the proposed PPT method for class prior estimation.

### 2.1. Implicit Feedback Recommendation

Let $\mathcal{U}$ and $\mathcal{I}$ denote the sets of users and items, respectively, and $\mathcal{X} = \mathcal{U} \times \mathcal{I}$ represent the set of all possible user-item interactions, where each sample is denoted by $x = (u, i)$. We introduce $Y$ as the binary indicator variable for positive feedback and designate $p_{\text{data}}$ as the underlying distribution governing all interactions. The objective is to construct a recommendation model, symbolized as $g$, that is proficient in discriminating between samples associated with positive and negative feedback. To quantify the performance of the model, we employ an error measure $\ell(\hat{y}, y)$, where $y$ signifies the ground truth feedback and $\hat{y}$ represents the model's estimated feedback. In line with existing literature (Saito et al., 2020; Luo et al., 2021), the ideal risk function is formulated as:

$$R_{\text{ideal}}(g) = \mathbb{E}_{p_{\text{data}}(x,y)} \left[ \ell(g(x), y) \right]. \tag{1}$$

The samples with negative feedback, i.e., $p_{\text{data}}(x|y = -1)$, is unavailable in ImplicitRec due to the MNF issue, rendering the ideal risk in (1) incalculable. Current approaches typically assume that unobserved samples are all negative, resorting to negative sampling on these samples to generate negative feedback. However, this assumption does not hold in real-world practice as the negatively sampled data inevitably contain a proportion of positive feedback. Assigning negative labels to them can lead to an unstable training process and compromise overall performance (Rendle et al., 2009; Saito, 2020).

### 2.2. Optimal Transport

Optimal transport (OT) is a mathematical framework designed to measure the discrepancy between two distributions by calculating the minimum transport cost, which has been used in many fundamental fields due to its empirical flexibility and theoretical rigor: computer vision (Courty et al., 2017; Xu et al., 2021; Shen et al., 2018). Initially, Monge (1781) first formulated it as finding an optimal mapping between two continuous distributions. However, this formulation had limitations in terms of the existence and uniqueness of solutions. A more practical formulation was later proposed by Kantorovich (2006), with an entropic regularizer for enabling fast computation via Sinkhorn algorithm (Altschuler et al., 2017). We provide its specification for comparing two empirical distributions in Definition 1.

**Definition 1.** *For empirical distributions $\alpha_{1:n}$ and $\beta_{1:m}$ with n and m samples, respectively, the Kantorovich problem aims to find an optimal plan $\boldsymbol{\pi} \in \mathbb{R}_+^{n \times m}$ that minimizes the*

---

[1]The term propensity means the likelihood that the sample would be observed. It is quite difficult to be estimated in the realm of ImplicitRec due to MNF, since the samples that are observed but disliked is unknown.

*transport cost* $\mathbb{W}(\alpha, \beta) \in \mathbb{R}$ *between* $\alpha$ *and* $\beta$. *Formally, the problem is defined as:*

$$
\begin{aligned}
\mathbb{W}(\alpha, \beta) &:= \min_{\boldsymbol{\pi} \in \Pi(\alpha, \beta)} \langle \mathbf{C}, \boldsymbol{\pi} \rangle, \\
\Pi(\alpha, \beta) &:= \left\{ \boldsymbol{\pi} \in \mathbb{R}_+^{n \times m} : \boldsymbol{\pi} \mathbf{1}_m = \mathbf{a}, \boldsymbol{\pi}^\top \mathbf{1}_n = \mathbf{b} \right\},
\end{aligned} \tag{2}
$$

*where* $\mathbf{C} \in \mathbb{R}_+^{n \times m}$ *denotes the sample-wise distance between* $\alpha$ *and* $\beta$. $\mathbf{1}_n$ *and* $\mathbf{1}_m$ *are column vectors filled with ones. The set* $\Pi$ *defines feasible transport plans.* $\mathbf{a}$ *and* $\mathbf{b}$ *specify the mass of units in* $\alpha$ *and* $\beta$.

Recent research in OT primarily follows two trajectories. The first aims to reduce the computational complexity of solving OT problems. While exact solutions can be obtained through linear programming algorithms, these come with cubic complexity in relation to the number of samples (Bonneel et al., 2011). To address this, various approximate algorithms for acceleration have been developed, such as the Sinkhorn and sliced OT algorithm (Altschuler et al., 2017) with quadratic and linear complexity, respectively. The second line of research focuses on modifying the transport problem to suit specific applications (Wang et al., 2025a;b). Examples include the Schrödinger bridge problem in generative modeling (Marino and Gerolin, 2020), the Gromov problem in graph matching (Xu et al., 2019), and the weak transport problem in causal inference (Wang et al., 2023).

## 3. Methodology

This section presents our proposed WeaklyRec framework. We first construct a surrogate estimator of the ideal risk (1) utilizing only positive and unlabeled samples. A thorough theoretical analysis is provided to expound upon its advantageous attributes. We then formulate the Progressive Proximal Transport (PPT) method to achieve automatic estimation of the class prior. The overall workflow of WeaklyRec is finally encapsulated for implementation.

### 3.1. Weakly Supervised Recommendation

Assuming that positive and negative samples arise from marginal distributions $p_\text{p}(x) = p_\text{data}(x|y = 1)$ and $p_\text{n}(x) = p_\text{data}(x|y = -1)$, respectively, the data distribution can be expressed as:

$$
p_\text{data}(x) = \kappa_\text{p} p_\text{p}(x) + \kappa_\text{n} p_\text{n}(x), \tag{3}
$$

where $\kappa_\text{p} = p(y = +1)$ and $\kappa_\text{n} = p(y = -1)$ are the class priors, satisfying $\kappa_\text{p} + \kappa_\text{n} = 1$. Decomposing the ideal risk in (1) yields:

$$
\begin{aligned}
R_\text{ideal}(g) &= \kappa_\text{p} \mathbb{E}_{p_\text{p}}[\ell(g(x), +1)] + \kappa_\text{n} \mathbb{E}_{p_\text{n}}[\ell(g(x), -1)] \\
&:= \kappa_\text{p} R_\text{p}(g) + \kappa_\text{n} R_\text{n}(g),
\end{aligned}
$$

where $R_\text{p}(g)$ and $R_\text{n}(g)$ denote the expected error on positive and negative samples, respectively. Since negative

samples are inaccessible due to MNF, a workaround involves leveraging the unlabeled data distribution, which is a mixture of positive and negative data distributions:

$$
p_\text{u}(x) = \kappa_\text{p} p_\text{p}(x) + \kappa_\text{n} p_\text{n}(x).
$$

On the basis of this, the negative risk $R_\text{n}(g)$ can be expressed in terms of the unlabeled and positive samples:

$$
\begin{aligned}
\kappa_\text{n} R_\text{n}(g) &= \kappa_\text{n} \mathbb{E}_{p_\text{n}}[\ell(g(x), -1)] \\
&= \kappa_\text{n} \int \ell(g(x), -1) p_\text{n}(x) \mathrm{d}x \\
&= \int \ell(g(x), -1) p_\text{u}(x) \mathrm{d}x \\
&\quad - \kappa_\text{p} \int \ell(g(x), -1) p_\text{p}(x) \mathrm{d}x \\
&= \mathbb{E}_{p_\text{u}}[\ell(g(x), -1)] - \kappa_\text{p} \mathbb{E}_{p_\text{p}}[\ell(g(x), -1)].
\end{aligned} \tag{4}
$$

Introducing $\tilde{\ell}(g(x)) = \ell(g(x), +1) - \ell(g(x), -1)$, we can eliminate the expectation over the negative samples in (1) to derive a surrogate of the ideal risk:

$$
R_\text{weak}(g) = \kappa_\text{p} \mathbb{E}_{p_\text{p}} \left[ \tilde{\ell}(g(x)) \right] + \mathbb{E}_{p_\text{u}} \left[ \ell(g(x), -1) \right], \tag{5}
$$

where subtracting the risk of misclassifying positive samples as negative compensates for the bias introduced by treating all unlabeled samples as negative. Consequently, the WeaklyRec risk excludes unavailable negative feedback, enabling empirical estimation using positive and unobserved samples:

$$
\hat{R}_\text{weak}(g) = \frac{\kappa_\text{p}}{n_\text{p}} \sum_{i=1}^{n_\text{p}} \tilde{\ell}(g(x_i^\text{p})) + \frac{1}{n_\text{u}} \sum_{i=1}^{n_\text{u}} \ell(g(x_i^\text{u}), -1). \tag{6}
$$

**Theoretical Justification.** We demonstrate that the *WeaklyRec estimator is unbiased relative to the ideal risk* enabling training and evaluation without reliance on negative samples (Theorem 1). Moreover, *adding more unlabeled samples consistently reduces estimation error* (Theorem 3), suggesting an opportunity to improve ImplicitRec performance. Thanks to the convexity of $\hat{R}_\text{weak}(g)$, *the convergence of WeaklyRec is guaranteed with bounded errors in mild conditions (Lemma 3)*. Detailed proofs are provided in Appendix B.

### 3.2. Progressive Proximal Transport

The estimator in (6) necessitates a predefined positive class prior $\kappa_\text{p}$, a.k.a., the proportion of positive samples concealed within unlabeled data. Although canonical positive-unlabeled learning scenarios often assume $\kappa_\text{p}$ to be known, such an assumption is untenable in ImplicitRec where the class prior is costly and difficult to obtain, thereby necessitating class prior estimation. However, the task is difficult

in ImplicitRec due to MNF. Existing methods are mainly based on partial alignment (du Plessis et al., 2015a; 2017) which minimizes the f-divergence between labeled and unlabeled distributions. However, these approaches suffer from an inherent positive bias (Sugiyama et al., 2022) and are ill suited for data based on sparse ids.

**Definition 2** (Proximal Transport, PT). *Suppose $\alpha_{1:n}$ and $\beta_{1:m}$ are empirical distributions corresponding to samples with positive and unlabeled feedback, respectively. Given a mass weight $w \in \mathbb{R}_+$, we aim to find a transport plan $\pi^* \in \mathbb{R}^{n \times m}$ minimizing the transport cost between $\alpha$ and $\beta$. Formally, the problem is defined as:*

$$\pi^*(\alpha, \beta; w) := \arg\min_{\pi \in \Pi(\alpha, \beta; w)} \langle \mathbf{C}, \pi \rangle,$$

$$\Pi(\alpha, \beta; w) := \{\pi \in \mathbb{R}_+^{n \times m} : \pi \mathbf{1}_m = \mathbf{a}, \pi^\top \mathbf{1}_n \leq w * \mathbf{b},$$

$$\mathbf{1}_n^\top \pi \mathbf{1}_m = 1\},$$

(7)

*where $\langle \mathbf{C}, \pi \rangle$ represents the transport cost; $\mathbf{C} \in \mathbb{R}_+^{n \times m}$ denotes the sample-wise distance[2] between $\alpha$ and $\beta$. $\mathbf{1}_n$ and $\mathbf{1}_m$ are column vectors filled with ones. The set $\Pi$ defines the feasible transport plans. $\mathbf{a}$ and $\mathbf{b}$ specify the mass of units in $\alpha$ and $\beta$, respectively, and are implemented as $\mathbf{1}_n/n$ and $\mathbf{1}_m/m$ under the assumption of a uniform mass distribution.*

To address this gap, we introduce the Progressive Proximal Transport (PPT) method, specifically designed for class prior estimation in the context of our WeaklyRec framework. The core concept of PPT is the proximal transport (PT) problem, as outlined in Definition 2. Unlike the traditional Kantorovich problem, our approach adjusts the mass of $\beta$ by a weight factor $w \geq 1$ and modifies the mass preservation equality to allow for non-matched samples in $\beta$. Figure 1 illustrates how the PT transport strategy evolves with different values of $w$. When $w = 1$, PT matches all samples in $\beta$, effectively reducing to the standard Kantorovich problem. As $w$ increases, PT allows each sample in $\beta$ to be matched with multiple samples in $\alpha$. As a result, the positively labeled samples $\alpha$ will be matched towards the nearest sub-distribution within $\beta$, driven by minimizing the transport cost in (7).

The proposed PPT approach leverages this property of PT to estimate the class prior $\kappa_p$ by progressively adjusting the weight $w$. Starting with a large initial value of $w$, the matched samples in $\beta$ are primarily positive, as they are expected to be closely aligned with $\alpha$. As $w$ decreases, more samples in $\beta$ are considered for matching, without significantly increasing the transport cost due to the substantial overlap between the positive samples labeled and unlabeled. Upon reaching a threshold $w^*$, the inclusion of negative samples becomes unavoidable, leading to a sharp increase in the transport cost, since the distributions of positive and

---
[2]Here, we calculate the sample-wise distance with the squared Euclidean metric in line with Altschuler et al. (2017).

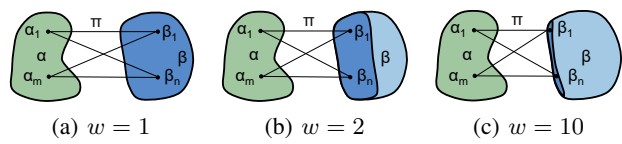

(a) $w = 1$     (b) $w = 2$     (c) $w = 10$

Figure 1: Overview of the PT transport plan under varying mass weight parameters $w$. The dark fields in $\beta$ demarcate the domain containing samples that are transported. Lines connecting $\alpha$ and $\beta$ signify the transport plans $\pi(\alpha, \beta; w)$.

---

**Algorithm 1** The computation workflow of WeaklyRec.

**Input**: a mini-batch of samples $\mathbf{x}$, recommender model $g$ with user and item embedding look-up table $\mathbf{U}$ and $\mathbf{V}$.
**Parameter**: $\eta$: learning rate; $w_{1:K}$: mass weights.
**Output**: updated embedding look-up tables $\mathbf{U}^\dagger$ and $\mathbf{V}^\dagger$.

1: $x_{1:n_p}^p, x_{1:n_u}^u \leftarrow \text{split}(\mathbf{x})$.
2: $\mathbf{u}_i^p, \mathbf{v}_i^p \leftarrow \text{look-up}(\mathbf{U}, \mathbf{V}, x_i^p), \mathbf{z}_i^p \leftarrow \text{concat}(\mathbf{u}_i^p, \mathbf{v}_i^p), i = 1 : n_p$.
3: $\mathbf{u}_j^u, \mathbf{v}_j^u \leftarrow \text{look-up}(\mathbf{U}, \mathbf{V}, x_j^u), \mathbf{z}_j^u \leftarrow \text{concat}(\mathbf{u}_j^u, \mathbf{v}_j^u), j = 1 : n_u$.
4: $c_{ij} \leftarrow \|\mathbf{z}_i^p - \mathbf{z}_j^u\|_2^2, \mathbf{C} \leftarrow [c_{ij}]_{i,j=1}^{n_p, n_u}$.
5: $\pi_k^* \leftarrow \arg\min_{\pi \in \Pi(\mathbf{z}_{1:n_p}^p, \mathbf{z}_{1:n_u}^u; w_k)} \langle \mathbf{C}, \pi \rangle, k = 1 : K$.
6: $\mathbb{W}_k \leftarrow \langle \mathbf{C}, \pi_k^* \rangle + \mathbf{KL}(\pi^\top \mathbf{1}_{n_p}, \mathbf{1}_{n_u}), k = 1 : K$.
7: $k^* \leftarrow \arg\min \mathbb{W}_{k=1:K}, \hat{\kappa} \leftarrow 1/w_{k^*}$.
8: $\hat{R}_{\text{weak}} \leftarrow \frac{\hat{\kappa}_p}{n_p} \sum_{i=1}^{n_p} \tilde{\ell}(g(\mathbf{u}_i^p, \mathbf{v}_i^p))_+ + \frac{1}{n_u} \sum_{j=1}^{n_u} \ell(g(\mathbf{u}_j^u, \mathbf{v}_j^u), -1)$.
9: $\mathbf{U}^\dagger \leftarrow \mathbf{U} - \eta \cdot \nabla_{\mathbf{U}} \hat{R}_{\text{weak}}, \mathbf{V}^\dagger \leftarrow \mathbf{V} - \eta \cdot \nabla_{\mathbf{V}} \hat{R}_{\text{weak}}$.

---

negative samples are heterogeneous. The class prior is then estimated as $\hat{\kappa}_p = 1/w^*$, justified by the fact that only approximately $1/w^*$ of the samples in $\beta$ are matched by PT as illustrated in Figure 1.

The PT developed in Definition 2 is inherently a linear programming problem and thus can be solved with well-established convex optimizers. Interestingly, PT can be viewed as an augmented Kantorovich problem (Xu et al., 2020), which enables the utilization of specialized optimal transport solvers, known for superior convergence rates and computational efficiency. A comprehensive analysis of this reformulation is provided in Appendix B.2.

### 3.3. Model Implementation

In this section, we delineate the implementation of WeaklyRec, employing matrix factorization as an exemplar recommender model. The computational workflow for a round of parameter updating is outlined in Algorithm 1.

Initially, we sample a mini-batch of samples, denoted as $\mathbf{x}$. This batch is subsequently partitioned into two subsets: positively-labeled samples $x_{1:n_p}^p$ and unlabeled sam-

Table 1: Recommendation performances in terms of NDCG@k on Yahoo! R3, Coat and KuaiRec.

| Dataset | Yahoo! R3 | | | Coat | | | KuaiRec | | |
|---|---|---|---|---|---|---|---|---|---|
| Metrics | NDCG@1 | NDCG@3 | NDCG@5 | NDCG@1 | NDCG@3 | NDCG@5 | NDCG@1 | NDCG@3 | NDCG@5 |
| RMF | $0.632_{\pm0.003}$ | $0.666_{\pm0.003}$ | $0.706_{\pm0.003}$ | $0.374_{\pm0.026}$ | $0.403_{\pm0.020}$ | $0.442_{\pm0.021}$ | $0.211_{\pm0.007}$ | $0.209_{\pm0.005}$ | $0.212_{\pm0.003}$ |
| CRMF | $0.633_{\pm0.004}$ | $0.667_{\pm0.003}$ | $0.706_{\pm0.003}$ | $0.377_{\pm0.022}$ | $0.401_{\pm0.018}$ | $0.445_{\pm0.023}$ | $0.212_{\pm0.005}$ | $0.218_{\pm0.004}$ | $0.222_{\pm0.002}$ |
| WMF | $0.707_{\pm0.003}$ | $0.737_{\pm0.003}$ | $0.772_{\pm0.002}$ | $0.389_{\pm0.025}$ | $0.425_{\pm0.023}$ | $0.466_{\pm0.027}$ | $0.226_{\pm0.009}$ | $0.244_{\pm0.008}$ | $0.261_{\pm0.008}$ |
| UBPR | $0.755_{\pm0.003}$ | $0.794_{\pm0.001}$ | $0.826_{\pm0.001}$ | $0.476_{\pm0.024}$ | $0.509_{\pm0.014}$ | $0.541_{\pm0.007}$ | $0.248_{\pm0.009}$ | $0.265_{\pm0.006}$ | $0.281_{\pm0.006}$ |
| CUBPR | $0.755_{\pm0.003}$ | $0.794_{\pm0.002}$ | $0.826_{\pm0.002}$ | $0.478_{\pm0.026}$ | $0.509_{\pm0.013}$ | $0.542_{\pm0.007}$ | $0.254_{\pm0.007}$ | $0.268_{\pm0.007}$ | $0.284_{\pm0.007}$ |
| UPL | $\underline{0.759}_{\pm0.003}$ | $\underline{0.798}_{\pm0.002}$ | $\underline{0.829}_{\pm0.002}$ | $\underline{0.512}_{\pm0.013}$ | $0.508_{\pm0.013}$ | $\underline{0.539}_{\pm0.012}$ | $0.250_{\pm0.006}$ | $0.265_{\pm0.006}$ | $0.281_{\pm0.005}$ |
| RecVAE | $0.650_{\pm0.002}$ | $0.683_{\pm0.004}$ | $0.719_{\pm0.005}$ | $0.465_{\pm0.021}$ | $0.499_{\pm0.017}$ | $0.545_{\pm0.009}$ | $0.422_{\pm0.006}$ | $0.434_{\pm0.005}$ | $0.448_{\pm0.007}$ |
| DCF | $0.762_{\pm0.003}$ | $0.796_{\pm0.003}$ | $0.826_{\pm0.002}$ | $0.544_{\pm0.0025}$ | $0.538_{\pm0.019}$ | $0.566_{\pm0.007}$ | $0.448_{\pm0.007}$ | $0.457_{\pm0.008}$ | $0.466_{\pm0.004}$ |
| ORACLE | $0.652_{\pm0.004}$ | $0.692_{\pm0.003}$ | $0.732_{\pm0.003}$ | $0.443_{\pm0.019}$ | $0.470_{\pm0.012}$ | $0.507_{\pm0.011}$ | $0.204_{\pm0.004}$ | $0.225_{\pm0.005}$ | $0.246_{\pm0.005}$ |
| CDR | $0.669_{\pm0.005}$ | $0.695_{\pm0.003}$ | $0.727_{\pm0.004}$ | $0.398_{\pm0.015}$ | $0.420_{\pm0.014}$ | $0.457_{\pm0.014}$ | $0.491_{\pm0.012}$ | $0.425_{\pm0.005}$ | $0.412_{\pm0.004}$ |
| SDR | $0.666_{\pm0.001}$ | $0.688_{\pm0.001}$ | $0.719_{\pm0.001}$ | $0.452_{\pm0.030}$ | $0.467_{\pm0.022}$ | $0.502_{\pm0.018}$ | $0.493_{\pm0.006}$ | $0.419_{\pm0.010}$ | $0.400_{\pm0.010}$ |
| UIDR | $0.679_{\pm0.030}$ | $0.705_{\pm0.032}$ | $0.737_{\pm0.033}$ | $0.382_{\pm0.015}$ | $0.400_{\pm0.011}$ | $0.436_{\pm0.014}$ | $\underline{0.493}_{\pm0.085}$ | $\underline{0.446}_{\pm0.060}$ | $\underline{0.435}_{\pm0.050}$ |
| WeaklyRec | $\mathbf{0.784}^{*}_{\pm0.003}$ | $\mathbf{0.814}^{*}_{\pm0.003}$ | $\mathbf{0.843}^{*}_{\pm0.002}$ | $\mathbf{0.552}^{*}_{\pm0.031}$ | $\mathbf{0.555}^{*}_{\pm0.019}$ | $\mathbf{0.589}^{*}_{\pm0.015}$ | $\mathbf{0.498}_{\pm0.008}$ | $\mathbf{0.486}_{\pm0.007}$ | $\mathbf{0.488}^{*}_{\pm0.004}$ |

*Note*: The results are reported in $\mathrm{mean}_{\pm\mathrm{std}}$ with 5 runs. The best and second best metrics are bolded and underlined, respectively. "*" marks the metrics that WeaklyRec surpasses the best baseline with p-value $< 0.05$ over paired samples t-test.

ples $x^{\mathrm{u}}_{1:n_{\mathrm{u}}}$ (step 1). Following this partitioning, we look up the embedding tables to generate user and item embeddings (steps 2-3). The sample-wise distances between these embeddings are then computed (step 4). For each candidate mass weight $w_k$, we calculate the transport strategy by solving the PT problem in (7) and calculate the associated transport cost (steps 5-6). Notably, the cost is augmented with a KL divergence term to facilitate comprehensive matching of the positive samples within the unlabeled set.

Upon identifying the optimal mass weight $w_{k^*}$ that minimizes the transport cost, we estimate the class prior as $\hat{\kappa} = 1/w_{k^*}$ (step 7). Subsequently, we employ $\hat{\kappa}$ to calculate the surrogate unbiased risk (5). Notably, while the surrogate risk (5) is inherently non-negative, its empirical approximation in (6) does not preserve this non-negativity. If $\ell$ is not upper-bounded, the empirical risk (6) may lack a lower bound, thereby risking model overfitting (Kiryo et al., 2017; Chaudhari and Shevade, 2012). This challenge is not exclusive to our framework; it has also been observed but not adequately addressed in prior methodologies within the ImplicitRec field (Saito et al., 2020; Saito, 2020). Therefore, we nullify the negative values of $\tilde{l}$ by setting them to zero, so that negative terms are excluded from gradient calculation (step 8). Finally, the embedding tables are updated using stochastic gradient descent (step 9).

# 4. Experiments

## 4.1. Experimental Setup

**Dataset.** We utilize three real-world datasets: **Yahoo! R3**, **Coat**, and **KuaiRec**. These datasets are selected since they uniquely provide negative feedback for evaluating model performance and include unbiased test sets that simulate production environments, aligning with established Implic-

itRec studies (Saito et al., 2020; Saito, 2020; Ren et al., 2023). In Yahoo! R3 and Coat, user-item pairs with ratings above 4 are labeled as positive, and the rest as negative. In KuaiRec, records with counts below two are considered negative, while the others are positive. Each dataset is chronologically split into training, validation, and testing sets in a ratio of 0.8:0.1:0.1.

**Baselines.** In line with Ren et al. (2023), the collection of baselines encompasses models as follows.

- **RecVAE** (Shenbin et al., 2020) and **DCF** (He et al., 2024) are representative heuristic ImplicitRec models. **WMF** (Hu et al., 2008), **RMF** (Saito et al., 2020) and the clipped **CRMF** (Saito et al., 2020) are *unbiased* ImplicitRec methods with point-wise error measures.

- **UBPR** (Saito, 2020), **CUBPR** (Saito, 2020) and **UPL** (Ren et al., 2023) are *unbiased* ImplicitRec methods with pair-wise error measures. **UPL** (Ren et al., 2023) integrates the point-wise and pair-wise error measures, which to our knowledge *is currently the state-of-the-art method tailored for ImplicitRec*.

- **Oracle** is the recommender model that is trained using the ideal loss, but suffers from data sparsity of available explicit feedback (Wang et al., 2022) and selection bias. **SDR** (Li et al., 2023c), **CDR** (Song et al., 2023) and **UIDR** (Li et al., 2024) correct this bias via propensity weighting, where propensities are heuristically estimated following Saito et al. (2020).

**Training Protocol.** We use matrix factorization as the primary recommendation model, following existing studies (Ren et al., 2023; Saito et al., 2020; Saito, 2020; Li et al., 2023c). Experiments are conducted with two Intel

Table 2: Recommendation performances in terms of Recall@k on Yahoo! R3, Coat and KuaiRec.

| Dataset | Yahoo! R3 | | | Coat | | | KuaiRec | | |
|---|---|---|---|---|---|---|---|---|---|
| Metrics | Recall@1 | Recall@3 | Recall@5 | Recall@1 | Recall@3 | Recall@5 | Recall@1 | Recall@3 | Recall@5 |
| RMF | $0.042_{\pm0.002}$ | $0.132_{\pm0.005}$ | $0.226_{\pm0.003}$ | $0.054_{\pm0.017}$ | $0.170_{\pm0.018}$ | $0.285_{\pm0.022}$ | $0.008_{\pm0.001}$ | $0.022_{\pm0.002}$ | $0.037_{\pm0.001}$ |
| CRMF | $0.042_{\pm0.002}$ | $0.132_{\pm0.004}$ | $0.226_{\pm0.004}$ | $0.055_{\pm0.014}$ | $0.168_{\pm0.018}$ | $0.289_{\pm0.030}$ | $0.009_{\pm0.002}$ | $0.029_{\pm0.003}$ | $0.047_{\pm0.001}$ |
| WMF | $0.094_{\pm0.002}$ | $0.203_{\pm0.003}$ | $0.288_{\pm0.002}$ | $0.061_{\pm0.011}$ | $0.192_{\pm0.025}$ | $0.311_{\pm0.035}$ | $0.015_{\pm0.003}$ | $0.058_{\pm0.007}$ | $0.104_{\pm0.008}$ |
| UBPR | $0.125_{\pm0.002}$ | $0.265_{\pm0.000}$ | $0.344_{\pm0.001}$ | $0.113_{\pm0.008}$ | $0.270_{\pm0.009}$ | $0.375_{\pm0.013}$ | $0.023_{\pm0.004}$ | $0.069_{\pm0.005}$ | $0.115_{\pm0.005}$ |
| CUBPR | $0.126_{\pm0.003}$ | $0.265_{\pm0.002}$ | $0.344_{\pm0.002}$ | $0.113_{\pm0.009}$ | $0.270_{\pm0.009}$ | $0.378_{\pm0.010}$ | $0.024_{\pm0.002}$ | $0.071_{\pm0.008}$ | $0.119_{\pm0.009}$ |
| UPL | $\underline{0.129}_{\pm0.002}$ | $\underline{0.268}_{\pm0.002}$ | $\underline{0.346}_{\pm0.003}$ | $\underline{0.118}_{\pm0.002}$ | $0.256_{\pm0.011}$ | $0.369_{\pm0.010}$ | $0.022_{\pm0.000}$ | $0.070_{\pm0.004}$ | $0.117_{\pm0.004}$ |
| RecVAE | $0.054_{\pm0.002}$ | $0.149_{\pm0.001}$ | $0.234_{\pm0.002}$ | $0.096_{\pm0.012}$ | $0.267_{\pm0.021}$ | $0.389_{\pm0.031}$ | $0.110_{\pm0.001}$ | $0.183_{\pm0.002}$ | $0.214_{\pm0.002}$ |
| DCF | $0.131_{\pm0.003}$ | $0.262_{\pm0.003}$ | $0.339_{\pm0.004}$ | $0.134_{\pm0.014}$ | $0.272_{\pm0.024}$ | $0.386_{\pm0.027}$ | $0.119_{\pm0.001}$ | $0.231_{\pm0.005}$ | $0.288_{\pm0.004}$ |
| ORACLE | $0.055_{\pm0.003}$ | $0.162_{\pm0.004}$ | $0.258_{\pm0.004}$ | $0.085_{\pm0.009}$ | $0.233_{\pm0.009}$ | $0.350_{\pm0.015}$ | $0.011_{\pm0.001}$ | $0.052_{\pm0.005}$ | $0.101_{\pm0.006}$ |
| CDR | $0.066_{\pm0.002}$ | $0.157_{\pm0.003}$ | $0.235_{\pm0.006}$ | $0.064_{\pm0.012}$ | $0.185_{\pm0.018}$ | $0.297_{\pm0.017}$ | $\underline{0.140}_{\pm0.004}$ | $0.191_{\pm0.002}$ | $0.207_{\pm0.003}$ |
| SDR | $0.061_{\pm0.001}$ | $0.149_{\pm0.003}$ | $0.223_{\pm0.002}$ | $0.085_{\pm0.017}$ | $0.224_{\pm0.014}$ | $0.337_{\pm0.013}$ | $0.138_{\pm0.002}$ | $0.184_{\pm0.007}$ | $0.191_{\pm0.010}$ |
| UIDR | $0.075_{\pm0.021}$ | $0.168_{\pm0.035}$ | $0.245_{\pm0.037}$ | $0.060_{\pm0.009}$ | $0.163_{\pm0.011}$ | $0.268_{\pm0.013}$ | $0.133_{\pm0.037}$ | $\underline{0.206}_{\pm0.038}$ | $\underline{0.235}_{\pm0.033}$ |
| WeaklyRec | $\mathbf{0.146}^*_{\pm0.002}$ | $\mathbf{0.280}^*_{\pm0.003}$ | $\mathbf{0.354}^*_{\pm0.002}$ | $\mathbf{0.131}_{\pm0.016}$ | $\mathbf{0.302}^*_{\pm0.019}$ | $\mathbf{0.425}^*_{\pm0.019}$ | $\mathbf{0.137}_{\pm0.002}$ | $\mathbf{0.245}^*_{\pm0.005}$ | $\mathbf{0.296}^*_{\pm0.004}$ |

*Note*: The results are reported in $\text{mean}_{\pm\text{std}}$ with 5 runs. The best and second best metrics are bolded and underlined, respectively. "*" marks the metrics that WeaklyRec surpasses the best baseline with p-value $< 0.05$ over paired samples t-test.

Xeon Platinum 8383C CPUs (2.70 GHz) and eight NVIDIA GeForce RTX 4090 GPU. We tune the learning rate in $\{0.005, 0.01, 0.05\}$, batch size in $\{256, 512, 1024, 2048\}$, and embedding size in $\{8, 16, 32\}$. All experiments are implemented in PyTorch using the Adam optimizer (Kingma and Ba, 2015) with early stopping (patience = 5). The candidate mass weights ($w_k$) are set to [5, 10, 20, 50, 100] which balances searching spectrum and efficiency.

**Evaluation Protocol.** In the evaluation phase, the negative feedback is employed for performance evaluation. Two commonly used metrics are adopted for evaluation: Top-$k$ recall (Recall@$k$) and Top-$k$ normalized discounted cumulative gain (NDCG@$k$). We report results on Recall@$\{1, 3, 5\}$ and NDCG@$\{1, 3, 5\}$. To perform a significance test, all experiments are repeated for 5 times.

### 4.2. Overall Performance

The evaluation of our proposed methods and the baselines is summarized using the NDCG in Table 2 and Recall metrics in Table 5. We outline the key findings as follows:

- Conventional point-wise baselines exhibit practical performance but have inherent limitations: the sensitivity to small propensity scores leads to large estimation variances. Techniques like CRMF address this by capping propensities below a certain threshold, which introduces a bias and compromises performance.

- The pair-wise baselines such as UBPR and CUBPR also rely on propensity scores and hence share the limitations of point-wise baselines but generally outperform them. This improvement may come from the pair-wise loss, which penalize lower ratings for unlabeled samples rather than fixing them as negatives. Among the baselines, the

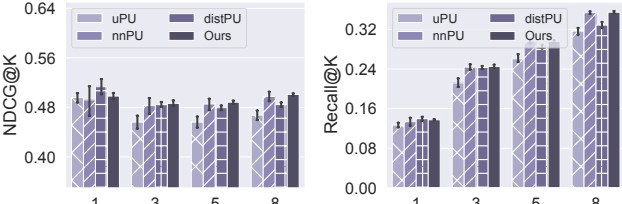

Figure 2: Varying risk estimator results on KuaiRec.

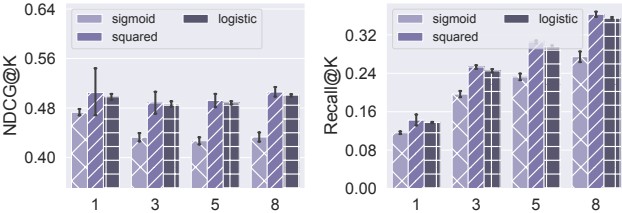

Figure 3: Varying error measure ($\ell$) results on KuaiRec.

UPL (Ren et al., 2023) approach, which combines point-wise and pair-wise strategies, is notably effective.

- Our WeaklyRec approach outperforms baselines across most datasets and metrics. This superiority may come from its unbiasedness and consistency without theoretical guarantees. Crucially, WeaklyRec operates independently of propensity score identification, avoiding issues related to small propensities and inaccurate propensity estimation in baselines (Saito, 2020; Saito et al., 2020).

### 4.3. Discussion on the WeaklyRec estimator

In this section, we investigate the performance of various error measures and weakly supervised learning estimators in the WeaklyRec framework.

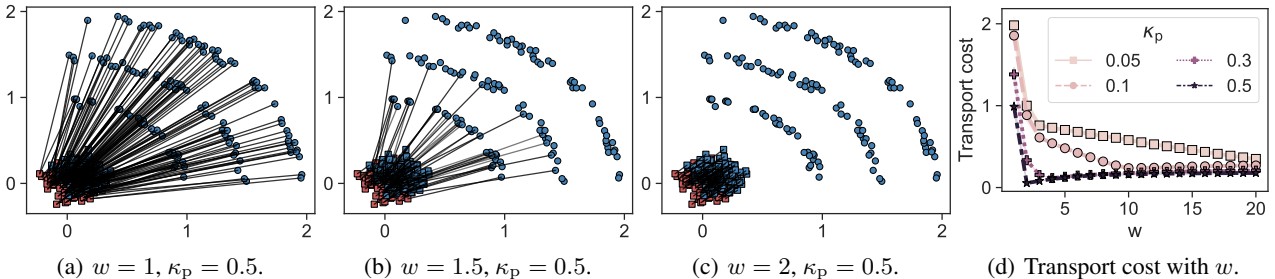

(a) $w = 1, \kappa_\mathrm{p} = 0.5$.      (b) $w = 1.5, \kappa_\mathrm{p} = 0.5$.      (c) $w = 2, \kappa_\mathrm{p} = 0.5$.      (d) Transport cost with $w$.

Figure 4: PT transport plan (a-c) and cost (d) with different $w$. Circular (rectangular) markers indicate positive (negative) samples. Blue (red) markers indicate labeled (unlabeled) samples.

Table 3: Varying fixed class prior results.

| $\hat{\kappa}_\mathrm{p}$ | $\mathbb{W}$ | $\Delta\mathbb{W}$ | NDCG@3 | $\Delta$NDCG@3 | NDCG@5 | $\Delta$NDCG@5 | Recall@3 | $\Delta$Recall@3 | Recall@5 | $\Delta$Recall@5 |
|---|---|---|---|---|---|---|---|---|---|---|
| 0.2 | 6.1143 | - | 0.4022 | - | 0.4232 | - | 0.2102 | - | 0.2732 | - |
| 0.005 | 5.5727 | 8.85%↓ | 0.4696 | 16.75%↑ | 0.4717 | 11.46%↑ | 0.2293 | 9.08%↑ | 0.2857 | 4.57%↑ |
| 0.01 | 5.5727 | 8.85%↓ | 0.4731 | 17.62%↑ | 0.4781 | 12.97%↑ | 0.2326 | 10/65%↑ | 0.2930 | 7.24%↑ |
| 0.02 | 5.5716 | 8.87%↓ | 0.4847 | 20.51%↑ | 0.4867 | 15.00%↑ | 0.2419 | 15.08%↑ | 0.2954 | 8.12%↑ |
| 0.05 | **5.5620** | 9.03%↓ | **0.4847** | 20.51%↑ | **0.4871** | 15.09%↑ | **0.2455** | 16.79%↑ | **0.2979** | 9.04%↑ |
| 0.1 | 5.5951 | 8.49%↓ | 0.4427 | 10.06%↑ | 0.4564 | 7.84%↑ | 0.2296 | 9.22%↑ | 0.2886 | 5.63%↑ |

*Note*: $\Delta$ denotes the relative deviation from the results with $\hat{\kappa}_\mathrm{p} = 0.2$.

- **Varying risk estimator results.** We incorporate three widely-used weakly supervised learning estimators into the WeaklyRec framework: uPU (du Plessis et al., 2014), nnPU (Kiryo et al., 2017), and distPU (Zhao et al., 2022). Our empirical results in Figure 2 indicate that all three estimators deliver commendable performance. Specifically, uPU, a seminal work in this area, offers theoretically unbiased risk estimates. However, it suffers from consequent overfitting due to the absence of a lower bound on the loss function. Subsequent nnPU and distPU address this issue by introducing non-negativity constraints, resulting in improved performance. However, they impose non-negativity constraints in an expectation sense, overlooking the role of individual non-negative terms within the expectation and thereby compromising sample efficiency. In contrast, our implementation advocates to adding non-negative constraint for each term individually, which enhances sample efficiency and marginally improves performance across six out of eight metrics.

- **Varying error measure results.** According to Theorem 4, the surrogate risk function (5) with a logistic error measure is convex, making it amenable to optimization via stochastic gradient methods. This property also extends to the squared loss error measure, which explains its comparable performance relative to the logistic measure in Figure 3. On the other hand, the sigmoid error measure renders the surrogate risk non-convex, which complicates optimization and causes suboptimal performance.

### 4.4. Discussion on the PPT Approach

Owing to missing negative feedback, the actual class prior $\kappa$ in unavailable. To showcase the efficacy of PPT in class prior estimation, we make two efforts as follows.

#### 4.4.1. NUMERICAL VERIFICATION

We conduct a case study using a simulated dataset with a predefined class prior $\kappa_\mathrm{p}$. Figure 4 (a-c) illustrates the dataset and transport strategies for different class prior estimates $w$. The key observations are summarized as follows:

- Different values of $w$ result in distinct matching strategies $\pi$. At $w = 1$, PT reduces to the standard Kantorovich problem, matching all unlabeled samples. With $w = 1.5$, PT matches positively labeled samples with concealed positives but also matches some negatives due to mass preservation, leading to high transport costs. Increasing $w$ to 2 allows partial matching of only concealed positive samples, rectifying the transport plan and minimizing transport cost. Further increases in $w$, as shown in Figure 4 (d), do not rectify the transport plan and instead increase transport costs.

- The behavior of $w$ proves suitable for estimating class priors. Figure 4 (d) shows that for a true class prior $\kappa_\mathrm{p} = 0.3$, transport cost decreases sharply towards about 0.2 at $w = 0.4$, then gradually increases. According to Section 4.4, PPT estimates the class prior to be between 0.25 and 0.33, closely matching the predefined $\kappa_\mathrm{p} = 0.3$. For other class priors of 0.05, 0.1, and 0.5, the minimum

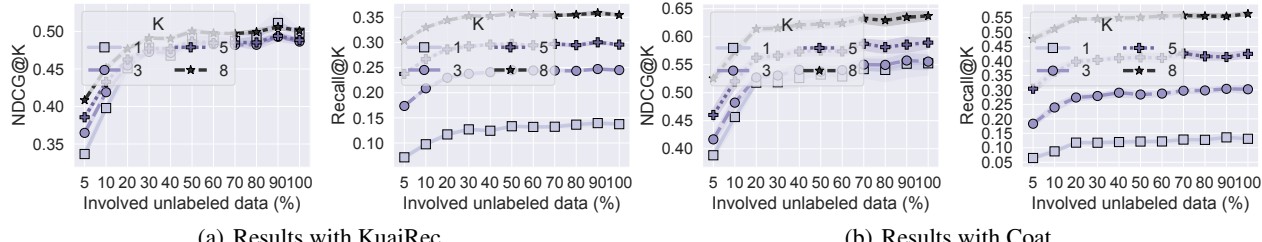

(a) Results with KuaiRec.             (b) Results with Coat.

Figure 5: Results with varying ratios of unlabeled data for model training. The colored lines and shadowed areas represent mean values and 90% confidence intervals, respectively.

transport costs correspond to $w$ values of 20, 10, and 2 respectively, supporting the relationship $\kappa_{\mathrm{p}} = 1/w$.

#### 4.4.2. IN-LOOP VERIFICATION

A reasonable hypothesis is that the class prior yielding the optimal performance likely corresponds to the ground truth, aligning well with both theoretical properties and practical requirements. Therefore, we firstly modify WeaklyRec to use fixed class prior estimates, $\hat{\kappa}$, and evaluate the resulting performance. Afterwards, we examine whether PPT's estimated class prior approximates the fixed prior that achieves the best performance. This analysis is conducted using the large-scale KuaiRec dataset, with key results summarized in Table 3 and analyzed below.

- Accurate class prior estimation is critical for WeaklyRec's performance. In Table 3, setting $\hat{\kappa}_{\mathrm{p}} = 0.2$ results in an NDCG@5 of approximately 0.423. Performance consistently improves as $\hat{\kappa}_{\mathrm{p}}$ decreases, reaching the peak of around 0.487 at $\hat{\kappa}_{\mathrm{p}} = 0.05$. Further increasing $\hat{\kappa}_{\mathrm{p}}$ to 0.1 causes an immediate drop. Thus, the optimal class prior estimate is approximately $\hat{\kappa}_{\mathrm{p}} = 0.05$.

- PPT effectively identifies the ground-truth class prior. Specifically, the PT cost is minimized at $\hat{\kappa}_{\mathrm{p}} = 0.05$ at 5.5620, which corresponds to the best overall performance in Table 3. Therefore, the PPT approach successfully identifies the class prior that optimizes performance.

#### 4.5. Scalability Analysis

According to Theorem 3, involving more unlabeled samples reduces the variance to approximate the ideal risk and improves performance. In this section, we investigate the performance of WeaklyRec given varying proportions of unlabeled samples for training, to substantiate the utility of unlabeled samples.

The results in Figure 5 show a consistent improvement in performance with increasing proportions of unlabeled

samples. For instance, using just 5% of unlabeled samples achieves an NDCG@1 of approximately 0.34 on the KuaiRec dataset, with performance peaking at around 0.5 when all unlabeled samples are included. These trends, which hold across other metrics and datasets, validate WeaklyRec's theoretical consistency. Additionally, the rate of performance improvement tends to plateau beyond a certain threshold, indicating that while it is possible to use all unlabeled data, a subset may be sufficient to balance performance and training cost.

## 5. Related Works

### 5.1. Learning from Weak Supervision

Weakly supervised learning refers to a type of machine learning paradigm where the training data is partially labeled or noisily labeled (Sugiyama et al., 2022). Unlike supervised learning wherein each data point has a corresponding accurate label, weak supervised learning operates on data where the labels may be imprecise, incomplete, or altogether missing for some instances. This approach is particularly useful when obtaining fully labeled data is expensive or difficult.

Positive-Unlabeled (PU) learning an important instance of weakly supervised learning, focusing on problems where the training set consists of labeled positive examples and unlabeled examples that could either be positive or negative, with applications in fundamental fields such as computer vision (Loghmani et al., 2020; Chapel et al., 2020) and text processing (Le et al., 2020; S. et al., 2022). The core challenge arises from the absence of negative labels, which impedes the application of traditional supervised learning models. There are two primary paradigms to tackle this challenge: re-weighting and pseudo-labeling. The re-weighting paradigm adjusts sample weights to achieve unbiased estimation of the ideal risk, exemplified by uPU (du Plessis et al., 2014). Subsequent modifications make uPU convex (du Plessis et al., 2015b) and non-negative (Kiryo et al., 2017; Chaudhari and Shevade, 2012; Zhang and Zuo, 2009), to accommodate with stochastic gradient optimizers. Recently, further refinements have been proposed to adapt

handle data imbalance (Su et al., 2021) and sample selection bias (Hammoudeh and Lowd, 2020). On the other hand, the pseudo-labeling paradigm identifies reliable negative and positive samples from the unlabeled data and then turns the problem into a supervised learning task. Methods differ in the techniques of pseudo labels assignment, such as graph neural network (Chaudhari and Shevade, 2012; Zhang and Zuo, 2009), generative models (Hou et al., 2018), confidence score (Northcutt et al., 2017), reinforcement learning (Luo et al., 2021) and clustering analysis (Gong et al., 2019).

### 5.2. Recommendation with Implicit Feedback

ImplicitRec aims to infer user preferences from user behaviors such as clicks or views rather than explicit comments.

Compared with explicit feedback, implicit feedback is characterised by missing negative feedback (MNF). Specifically, the data involves some positive feedback and unlabeled data. The absence of negative feedback makes it difficult to differentiate between items that are disliked and those simply unexposed to the user. Current approaches treating unlabeled data as negative can lead to biased and inaccurate recommendations (Zhou et al., 2022; Kang and McAuley, 2018; Wang et al., 2022; Li et al., 2023c). An exemplar approach to handle this issue is the weighted matrix factorization (Hu et al., 2008), which heuristically downweights the unobserved user-item samples versus the observed ones. ExpoMF (Liang et al., 2016) further extends it to a generative formulation. However, these methods are often heuristic and lack rigorous theoretical underpinning, indicating a need for further exploration of this line of works.

Recent advancement further considers sample selection bias in the positively labeled data. Specifically, the observed positive feedback is not a random sample of a user's true preferences; rather, it is influenced by various factors such as item popularity and visibility (Lim et al., 2015; Yang et al., 2018). This non-random sampling can distort the true underlying preference structure, resulting in inaccurate recommendations (Zhou et al., 2021; Gupta et al., 2023; Togashi et al., 2021). Current efforts mainly employ inverse propensity weighting (IPW) to correct such selection bias. It re-weights each observed sample using a propensity score, which theoretically ensures an unbiased modeling of user-item preferences (Wang et al., 2022; Li et al., 2023c;a). In practice, methods mainly differ in terms of the ways to estimate propensity, represented by employing the item popularity (Saito et al., 2020; Saito, 2020) and constructing a propensity estimator (Zhu et al., 2020; Lee et al., 2022; Xiao et al., 2024). Despite its theoretical merits, IPW faces several practical challenges: (i) the propensity estimation is often inaccurate (Li et al., 2023c), exacerbated by the absence of negative feedback; (ii) the estimators are sensitive to small propensity scores (Li et al., 2023a), complicating

the training process; and (iii) the non-negativity constraint, essential for model convergence, is often violated for the sake of theoretical unbiasedness (Saito et al., 2020; Saito, 2020). These limitations are not only theoretically complex but also empirically restrict the application of IPW-based approaches in ImplicitRec.

Recently, PU learning has also been incorporated for enhancing ImplicitRec (Togashi et al., 2021; Zhou et al., 2021). For example, Zhou et al. (2021) develop an adversarial approach using PU loss for discriminator training, Togashi et al. (2021) introduce a density-ratio-based method that leverages PU learning for density-ratio estimation. Our contributions are distinct from them: we are the first to integrate PU learning to enhance the binary classification objective for ImplicitRec and propose a class prior estimation method based on optimal transport, a significant contribution given the complexity of this issue in PU learning.

## 6. Conclusions

In conclusion, the challenge of missing negative samples has long been a significant obstacle in ImplicitRec. To handle this issue, we develop WeaklyRec, a model-agnostic framework that adapts principles from weakly supervised learning to the domain of ImplicitRec. A cornerstone of our approach is the PPT approach designed to estimate the class prior by minimizing the proximal transport cost between positive and unlabeled samples. Extensive evaluations on real-world datasets confirm the efficacy of our approach and support the theoretical analysis empirically.

**Limitation & Future works** This study adheres to the setting of ID-based recommendation research and does not consider user profiles. Future work could extend this framework to more content-rich scenarios. Additionally, the scalability of WeaklyRec highlights the potential of using implicit feedback to enhance recommenders. Therefore, how to harmonize explicit and implicit feedback presents a promising direction for further research.

## Acknowledgements

This work is supported by National Key R&D Program of China (2022ZD0160300) and the NSF China (No. 62276004, 623B2002).

## Impact Statement

This paper presents work whose goal is to advance the field of Machine Learning. There are many potential societal consequences of our work, none which we feel must be specifically highlighted here.

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

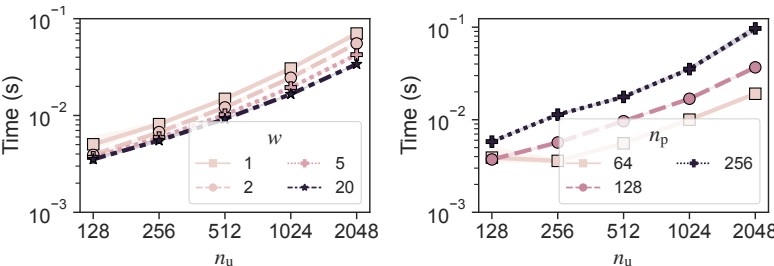

Figure 6: Running time of solving the PT problem. By default we set $n_\mathrm{p} = 128$ and $w = 10$.

## A. Additional Experimental Results

### A.1. Dataset Description

In this section, we present a brief introduction to the datasets involved in this study.

- **Yahoo! R3** contains 311,704 biased ratings for training, which involve $15,400$ users and $1,000$ items. Additionally, 5400 users rate 10 randomly selected items, yielding $54,000$ unbiased ratings for evaluation. The user-item pairs with rating greater than 4 are seen as positive, and others are viewed as negative.

- **Coat** is a public dataset which consists of 290 users and 300 items; each user subjectively selects 24 items to rate based on their preference, yielding 6,960 biased ratings in the training set. Additionally, each user rates 16 items that are randomly selected, yielding 4,640 unbiased ratings for model evaluation. The user-item pairs with rating greater than 4 are seen as positive, and others are viewed as negative.

- **KuaiRec** is a public large-scale industrial dataset, which consists of 4,676,570 video watching ratio records from 1,411 users for 3,327 videos. The user-item pairs with ratings less than two are viewed as negative, and otherwise are viewed as positive. The records less than two are viewed as negative feedback, and otherwise are viewed as negative feedback.

### A.2. Discussion on Complexity

Concerns may arise regarding the computational overhead introduced by WeaklyRec, given that it incorporates a linear programming procedure—the solving of the PT problem—in each iteration of model training. These concerns are particularly pertinent in the context of large batch sizes commonly employed in training recommender models, as this could render the PT problem large-scale and computationally intensive. To address these concerns, we empirically evaluate the time required for solving the PT problem under various settings, as depicted in Figure 6. We summarize two key observations as follows.

- The computational complexity of solving the PT problem is not prohibitive in our specific application. For instance, even when matching as many as 4096 unlabeled samples, the problem can be solved within 0.1 seconds. This relatively low computational burden is primarily due to the scarcity of positively labeled samples. In contrast to standard optimal transport (OT) scenarios where two distributions have a similar number of samples, the PT problem in our case involves matching $n_\mathrm{u}$ unlabeled samples with a significantly smaller number of $n_\mathrm{p}$ positive samples. Given that $n_\mathrm{p} << n_\mathrm{u}$, the computational cost is substantially lower than in traditional OT applications.

- The computational complexity is also influenced by the value of the mass weight $w$. Specifically, as $w$ decreases, more unlabeled samples are compelled to engage in matching due to the mass-preserving constraint, thereby increasing the computational cost. When $w = 1$, the PT problem reduces to the canonical Kantorovich problem. However, given that the class prior is generally small in ImplicitRec applications, candidate values for $w$ can be set to relatively large values, which further reduces the complexity of solving PT in WeaklyRec.

### A.3. Hyperparameter Sensitivity

We perform a comprehensive sensitivity analysis to examine the influence of four critical hyperparameters, namely batch size, learning rate, and embedding dimension, on the performance of the WeaklyRec model using the coat and Kuairec datasets. The results of these experiments are illustrated in Figure 8 and 7, respectively.

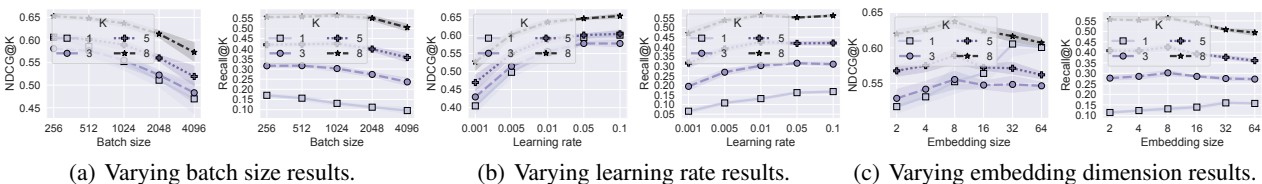

(a) Varying batch size results.     (b) Varying learning rate results.     (c) Varying embedding dimension results.

Figure 7: Parameter sensitivity studies of the WeaklyRec framework on Coat. Different colors indicate ranking metrics given different numbers of top candidates (K).

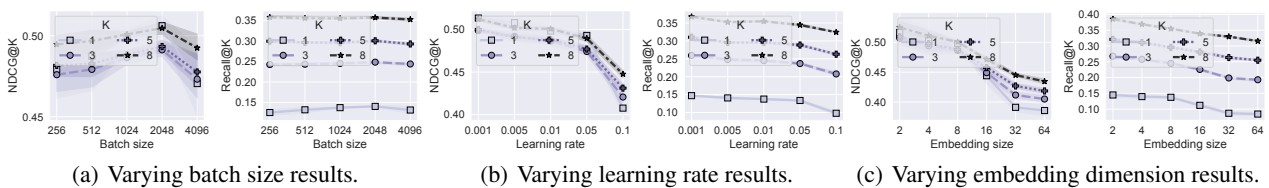

(a) Varying batch size results.     (b) Varying learning rate results.     (c) Varying embedding dimension results.

Figure 8: Parameter sensitivity studies of the WeaklyRec framework on Kuairec. Different colors indicate ranking metrics given different numbers of top candidates (K).

Initially, we examine the effects of varying batch sizes on two distinct datasets. Interestingly, as the batch size increases, metrics such as NDCG@K and Recall@K generally show a decreasing trend. However, the KuaiRec dataset exhibits an initial increase followed by a decrease in these performance measures. This fluctuation can potentially be attributed to the interplay between the inherent characteristics of the PPT algorithm, which is integral to our model architecture, and the properties of the datasets in question, as well as their batch sizes. Specifically, smaller batch sizes may lead the PPT algorithm to experience generalization issues, adversely impacting the model's overall performance. Conversely, larger batch sizes could introduce an over-smoothing problem, thereby reducing performance. These observations are further corroborated by data from the Coat and KuaiRec datasets in the batch size intervals of $[256, 4096]$ and $[2048, 4096]$ respectively. As for the interval $[256, 2048]$ in the KuaiRec dataset, this behavior can be explained by the dataset's size and inherent properties. Given the large size of the KuaiRec dataset, which primarily focuses on video clips, there is likely a significant level of noise. Furthermore, the dataset's user base, representing a variety of socio-economic strata, tends to rapidly dismiss recommended items. Consequently, smaller batch sizes may make the model more susceptible to noise.

Subsequently, we tuned the learning rate and observe divergent trends in performance across the Coat and KuaiRec datasets. Specifically, an increase in the learning rate leads to an improvement in performance metrics on the Coat dataset, while causing a decline on the KuaiRec dataset. This differential behavior can largely be attributed to the contrasting sizes of the two datasets. The KuaiRec dataset is considerably larger than the Coat dataset, and this disparity likely influences the performance outcomes. In larger datasets like KuaiRec, the presence of noise becomes more prevalent, which may adversely affect model performance when higher learning rates are employed.

Additionally, we investigate the influence of varying the embedding dimension, which ranges from 2 to 64. We observe that the model performance initially improves, only to subsequently decline. This fluctuating pattern can be linked to the inherent properties of the PPT algorithm integrated into our model. PPT computations are executed in the embedding space, and an overly restricted dimensionality may lack the capacity to accurately capture the data's complex manifold, leading to a degradation in model performance. On the other hand, an excessively large embedding dimension risks introducing redundancy and noise, thereby undermining the effectiveness of the optimal transport problem solution, which in turn may not reflect the true data manifold. For the KuaiRec dataset, a monotonically decreasing performance trend is evident. This observation lends further credence to our assertion that the KuaiRec dataset likely contains significant noise. Such noise interferes with the accurate representation of the true data manifold within the PPT algorithm, thereby adversely affecting the model performance.

In a broad range of hyperparameter settings, the WeaklyRec model consistently outperforms a majority of baseline models, further substantiating its superior performance characteristics. Specifically, even under varying conditions such as different learning rates, batch sizes, and embedding dimensions, the WeaklyRec model still excels or at least matches other commonly-

Table 4: Recommendation performances in terms of NDCG@k on Yahoo! R3, Coat and KuaiRec.

| Dataset | Yahoo! R3 | | | Coat | | | KuaiRec | | |
|---|---|---|---|---|---|---|---|---|---|
| Metrics | NDGC@1 | NDGC@3 | NDGC@5 | NDGC@1 | NDGC@3 | NDGC@5 | NDGC@1 | NDGC@3 | NDGC@5 |
| RMF | $0.632_{\pm 0.002}$ | $0.667_{\pm 0.003}$ | $0.707_{\pm 0.002}$ | $0.370_{\pm 0.022}$ | $0.394_{\pm 0.021}$ | $0.433_{\pm 0.021}$ | $0.209_{\pm 0.006}$ | $0.211_{\pm 0.006}$ | $0.215_{\pm 0.005}$ |
| CRMF | $0.633_{\pm 0.003}$ | $0.667_{\pm 0.002}$ | $0.706_{\pm 0.003}$ | $0.377_{\pm 0.019}$ | $0.400_{\pm 0.016}$ | $0.439_{\pm 0.019}$ | $0.211_{\pm 0.007}$ | $0.217_{\pm 0.006}$ | $0.222_{\pm 0.005}$ |
| WMF | $0.679_{\pm 0.030}$ | $0.710_{\pm 0.029}$ | $0.745_{\pm 0.028}$ | $0.402_{\pm 0.025}$ | $0.427_{\pm 0.018}$ | $0.465_{\pm 0.020}$ | $0.246_{\pm 0.049}$ | $0.256_{\pm 0.042}$ | $0.270_{\pm 0.043}$ |
| UBPR | $0.757_{\pm 0.003}$ | $0.795_{\pm 0.001}$ | $0.826_{\pm 0.002}$ | $0.459_{\pm 0.028}$ | $0.493_{\pm 0.021}$ | $0.530_{\pm 0.014}$ | $0.255_{\pm 0.010}$ | $0.272_{\pm 0.010}$ | $0.290_{\pm 0.011}$ |
| CUBPR | $0.756_{\pm 0.003}$ | $0.795_{\pm 0.002}$ | $0.826_{\pm 0.002}$ | $0.460_{\pm 0.030}$ | $0.493_{\pm 0.020}$ | $0.531_{\pm 0.015}$ | $0.257_{\pm 0.007}$ | $0.273_{\pm 0.009}$ | $0.292_{\pm 0.010}$ |
| UPL | $\underline{0.760}_{\pm 0.004}$ | $\underline{0.798}_{\pm 0.002}$ | $\underline{0.829}_{\pm 0.002}$ | $\underline{0.508}_{\pm 0.028}$ | $\underline{0.511}_{\pm 0.012}$ | $\underline{0.540}_{\pm 0.010}$ | $\underline{0.281}_{\pm 0.034}$ | $\underline{0.297}_{\pm 0.034}$ | $\underline{0.315}_{\pm 0.036}$ |
| ORACLE | $0.654_{\pm 0.003}$ | $0.693_{\pm 0.003}$ | $0.734_{\pm 0.003}$ | $0.425_{\pm 0.026}$ | $0.457_{\pm 0.019}$ | $0.496_{\pm 0.017}$ | $0.212_{\pm 0.011}$ | $0.231_{\pm 0.008}$ | $0.251_{\pm 0.008}$ |
| WeaklyRec | $\mathbf{0.787}_{\pm 0.004}$ | $\mathbf{0.817}_{\pm 0.003}$ | $\mathbf{0.845}_{\pm 0.003}$ | $\mathbf{0.579}_{\pm 0.035}$ | $\mathbf{0.567}_{\pm 0.019}$ | $\mathbf{0.597}_{\pm 0.015}$ | $\mathbf{0.489}_{\pm 0.013}$ | $\mathbf{0.480}_{\pm 0.009}$ | $\mathbf{0.483}_{\pm 0.007}$ |

*Note*: The results are reported in $\mathrm{mean}_{\pm \mathrm{std}}$ with 5 runs. The best and second best metrics are bolded and underlined, respectively. "*" marks the metrics that WeaklyRec surpasses the best baseline with p-value $< 0.05$ over paired samples t-test.

used recommendation algorithms in multiple evaluation metrics. This further confirms the robustness and versatility of the WeaklyRec model in practical recommendation system applications. Moreover, by conducting more fine-grained hyperparameter tuning, the WeaklyRec model has the potential to achieve even better performance than reported in the main text. This implies that, after more precise hyperparameter search and validation, the WeaklyRec model can not only maintain its advantage over baseline models but also possibly reach higher performance levels in specific application scenarios or datasets. This further highlights the superior and adaptive nature of the WeaklyRec model, making it worth further exploration in real-world applications.

### A.4. Supplementary Numerical Examples

To validate the efficacy of PPT in class prior estimation, we include a case study using a numerically simulated dataset with predefined (and known) class priors. In the main text, we only present the matching strategy on the simulated case wherein $\kappa_{\mathrm{p}} = 0.5$. Nevertheless, it is essential to consider other simulation conditions to back up the utility of the proposed approach.

To achieve this, we extend our numerical verification by considering a range of class priors $\kappa_{\mathrm{p}} = 0.05, 0.1, 0.3, 0.5$. We visualize the resulting transport strategies for different mass weights $w$. The key insights are consistent with those presented in the main text. The transport strategies are optimized when $w = 1/\kappa_{\mathrm{p}}$, at which point the transport cost reaches its minimum, as illustrated in Figure 4. These additional numerical scenarios further corroborate the effectiveness of the Progressive Proximal Transport (PPT) method in accurately estimating the class prior, thereby reinforcing the validity of our WeaklyRec framework.

### A.5. Performance with different backbone

In the main text, we implement the recommendation model $g$ as MF, since it is a widely used model to evaluate the efficacy of learning objectives in recommendation system (Li et al., 2023c;a), especially in recent ImplicitRec researches (Ren et al., 2023; Lee et al., 2022; Zhu et al., 2020; Saito et al., 2020). Nevertheless, we agree that it is interesting to investigate the performance of different ImplicitRec estimators in different backbones to test their generality. Therefore, we select NCF as our new backbone and summarize the experimental results in Table 4 and Table 5, which showcase trends similar to the results on MF.

## B. Theoretical Justification

### B.1. Formal Analysis on the Surrogate Risk

**Theorem 1.** *Given that the ideal risk of the recommender $g$ defined over all samples is $R_{\mathrm{ideal}}(g)$, the surrogate risk $\hat{R}_{\mathrm{weak}}(g)$ as defined in (6) is an unbiased estimator of $R_{\mathrm{ideal}}(g)$:*

$$\mathbb{E}_{\mathcal{X}}\left[\hat{R}_{\mathrm{weak}}(g)\right] = R_{\mathrm{ideal}}(g).$$

*Proof.* Let $\mathcal{X}$ be the empirical distribution of samples for training, with positive samples $\mathcal{X}_{\mathrm{p}}$ and unlabeled samples $\mathcal{X}_{\mathrm{u}}$.

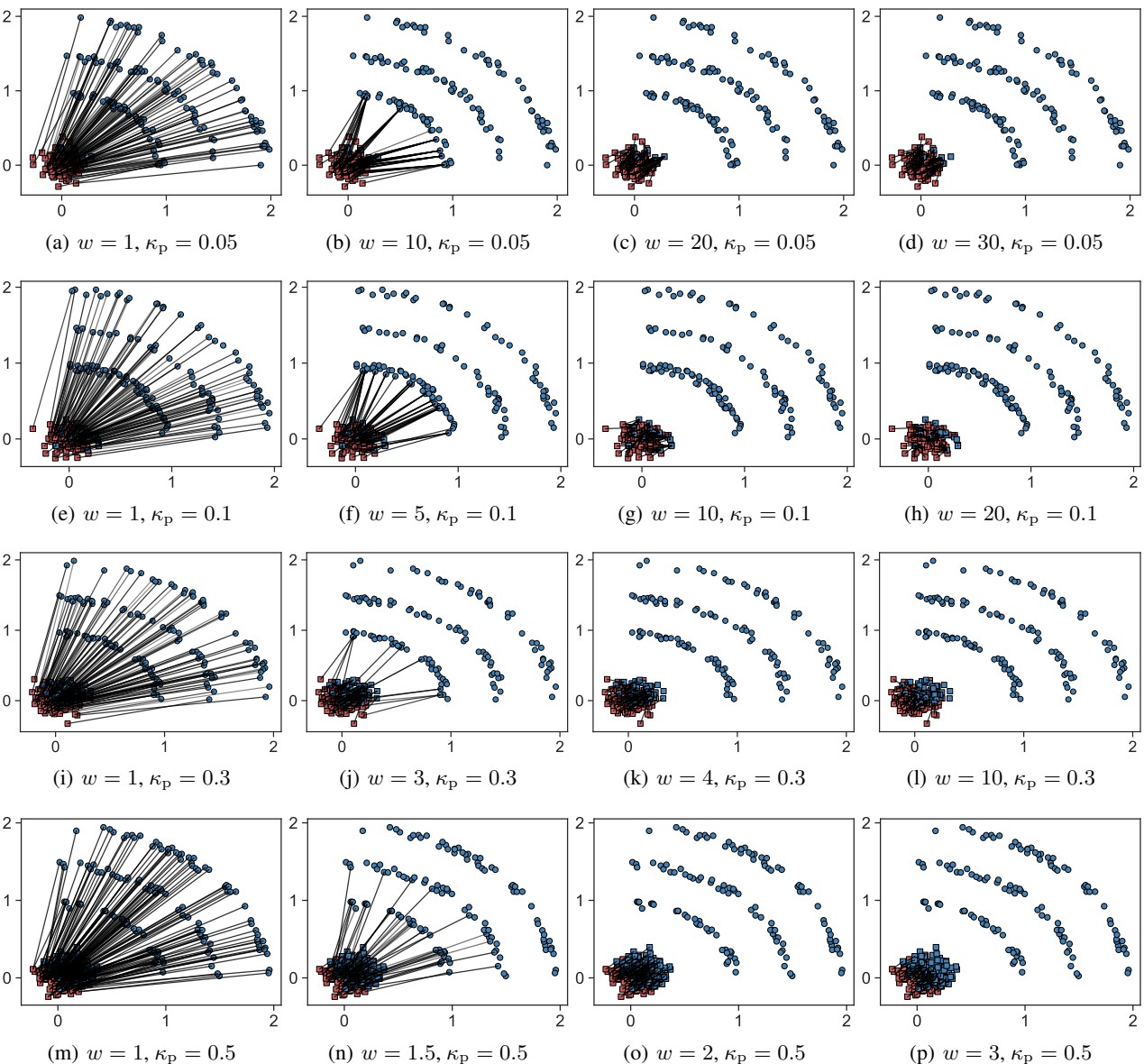

Figure 9: Overview of the PT transport plan given different weights of mass $w$ and ground truth positive priors $\kappa_{\mathrm{p}}$. The ground-truth positive and negative feedbacks are denoted by circular and rectangular markers, respectively. The labeled and unlabeled intersections are indicated in red and blue, respectively. The best transport strategies, which minimize transport costs for different values of $\kappa_{\mathrm{p}}$, are illustrated in subfigures (c, g, k, o), characterized by $1/w \approx \kappa_{\mathrm{p}}$.

Table 5: Recommendation performances in terms of Recall@k on Yahoo! R3, Coat and KuaiRec.

| Dataset | Yahoo! R3 | | | Coat | | | KuaiRec | | |
|---|---|---|---|---|---|---|---|---|---|
| Metrics | Recall@1 | Recall@3 | Recall@5 | Recall@1 | Recall@3 | Recall@5 | Recall@1 | Recall@3 | Recall@5 |
| RMF | $0.042_{\pm 0.002}$ | $0.133_{\pm 0.004}$ | $0.226_{\pm 0.003}$ | $0.054_{\pm 0.013}$ | $0.158_{\pm 0.022}$ | $0.270_{\pm 0.026}$ | $0.009_{\pm 0.001}$ | $0.025_{\pm 0.005}$ | $0.042_{\pm 0.006}$ |
| CRMF | $0.042_{\pm 0.002}$ | $0.132_{\pm 0.003}$ | $0.226_{\pm 0.004}$ | $0.056_{\pm 0.011}$ | $0.167_{\pm 0.016}$ | $0.277_{\pm 0.024}$ | $0.009_{\pm 0.002}$ | $0.030_{\pm 0.004}$ | $0.049_{\pm 0.005}$ |
| WMF | $0.074_{\pm 0.021}$ | $0.175_{\pm 0.030}$ | $0.260_{\pm 0.029}$ | $0.066_{\pm 0.013}$ | $0.192_{\pm 0.019}$ | $0.306_{\pm 0.025}$ | $0.023_{\pm 0.020}$ | $0.066_{\pm 0.033}$ | $0.108_{\pm 0.044}$ |
| UBPR | $0.126_{\pm 0.002}$ | $0.265_{\pm 0.001}$ | $0.344_{\pm 0.002}$ | $0.100_{\pm 0.017}$ | $0.254_{\pm 0.020}$ | $0.373_{\pm 0.013}$ | $0.026_{\pm 0.004}$ | $0.078_{\pm 0.011}$ | $0.128_{\pm 0.015}$ |
| CUBPR | $0.126_{\pm 0.002}$ | $0.265_{\pm 0.002}$ | $0.343_{\pm 0.003}$ | $0.100_{\pm 0.018}$ | $0.254_{\pm 0.020}$ | $0.374_{\pm 0.012}$ | $0.026_{\pm 0.003}$ | $0.078_{\pm 0.010}$ | $0.130_{\pm 0.014}$ |
| UPL | $\underline{0.129}_{\pm 0.004}$ | $\underline{0.267}_{\pm 0.003}$ | $\underline{0.345}_{\pm 0.002}$ | $\underline{0.117}_{\pm 0.012}$ | $\underline{0.261}_{\pm 0.011}$ | $\underline{0.367}_{\pm 0.010}$ | $\underline{0.033}_{\pm 0.012}$ | $\underline{0.096}_{\pm 0.028}$ | $\underline{0.153}_{\pm 0.039}$ |
| ORACLE | $0.056_{\pm 0.002}$ | $0.163_{\pm 0.003}$ | $0.260_{\pm 0.003}$ | $0.078_{\pm 0.011}$ | $0.221_{\pm 0.019}$ | $0.341_{\pm 0.021}$ | $0.014_{\pm 0.004}$ | $0.054_{\pm 0.006}$ | $0.102_{\pm 0.008}$ |
| WeaklyRec | $\mathbf{0.148}_{\pm 0.003}$ | $\mathbf{0.282}_{\pm 0.004}$ | $\mathbf{0.356}_{\pm 0.003}$ | $\mathbf{0.149}_{\pm 0.023}$ | $\mathbf{0.308}_{\pm 0.015}$ | $\mathbf{0.423}_{\pm 0.016}$ | $\mathbf{0.133}_{\pm 0.007}$ | $\mathbf{0.245}_{\pm 0.005}$ | $\mathbf{0.296}_{\pm 0.003}$ |

*Note*: The results are reported in $\mathrm{mean}_{\pm \mathrm{std}}$ with 5 runs. The best and second best metrics are bolded and underlined, respectively. "*" marks the metrics that WeaklyRec surpasses the best baseline with p-value $< 0.05$ over paired samples t-test.

According to the formulation of WeaklyRec in (6), the expectations regarding different training dataset can be formulated as follow:

$$
\begin{aligned}
\mathbb{E}_{\mathcal{X}}\left[\hat{R}_{\mathrm{weak}}(g)\right] &= \mathbb{E}_{\mathcal{X}_{\mathrm{p}}}\left[\frac{\kappa_{\mathrm{p}}}{n_{\mathrm{p}}}\sum_{i=1}^{n_{\mathrm{p}}}\tilde{\ell}(g(x_i^{\mathrm{p}}))\right] \\
&+ \mathbb{E}_{\mathcal{X}_{\mathrm{u}}}\left[\frac{1}{n_{\mathrm{u}}}\sum_{i=1}^{n_{\mathrm{u}}}\ell(g(x_i^{\mathrm{u}}), -1)\right]
\end{aligned}
\tag{8}
$$

For the first term, we have

$$
\begin{aligned}
\mathbb{E}_{\mathcal{X}_{\mathrm{p}}}\left[\frac{\kappa_{\mathrm{p}}}{n_{\mathrm{p}}}\sum_{i=1}^{n_{\mathrm{p}}}\tilde{\ell}(g(x_i^{\mathrm{p}}))\right] &= \frac{\kappa_{\mathrm{p}}}{n_{\mathrm{p}}}\sum_{i=1}^{n_{\mathrm{p}}}\mathbb{E}_{\mathcal{X}_{\mathrm{p}}}\left[\tilde{\ell}(g(x_i^{\mathrm{p}}))\right] \\
&= \frac{\kappa_{\mathrm{p}}}{n_{\mathrm{p}}}\sum_{i=1}^{n_{\mathrm{p}}}\mathbb{E}_{x_i^{\mathrm{p}}}\left[\tilde{\ell}(g(x_i^{\mathrm{p}}))\right] \\
&= \frac{\kappa_{\mathrm{p}}}{n_{\mathrm{p}}}\sum_{i=1}^{n_{\mathrm{p}}}\mathbb{E}_{p_{\mathrm{p}}}\left[\tilde{\ell}(g(x))\right] \\
&= \kappa_{\mathrm{p}}\mathbb{E}_{p_{\mathrm{p}}}\left[\tilde{\ell}(g(x^{\mathrm{p}}))\right]
\end{aligned}
\tag{9}
$$

For the second term, similarly we have

$$
\mathbb{E}_{\mathcal{X}_{\mathrm{u}}}\left[\frac{1}{n_{\mathrm{u}}}\sum_{i=1}^{n_{\mathrm{u}}}\ell(g(x_i^{\mathrm{u}}), -1)\right] = \mathbb{E}_{p_{\mathrm{u}}}\left[\ell(g(x), -1)\right].
\tag{10}
$$

Combining (8)-(10), we have

$$
\begin{aligned}
\mathbb{E}_{\mathcal{X}}\left[\hat{R}_{\mathrm{weak}}(g)\right] &= \kappa_{\mathrm{p}}\mathbb{E}_{p_{\mathrm{p}}}\left[\tilde{\ell}(g(x))\right] + \mathbb{E}_{p_{\mathrm{u}}}\left[\ell(g(x), -1)\right] \\
&= \kappa_{\mathrm{p}}\mathbb{E}_{p_{\mathrm{p}}}\left[\ell(g(x), +1)\right] \\
&+ (1 - \kappa_{\mathrm{p}})\mathbb{E}_{p_{\mathrm{n}}}\left[\ell(g(x), -1)\right] \\
&= R_{\mathrm{ideal}}(g).
\end{aligned}
\tag{11}
$$

$\square$

**Lemma 1.** *Let $\mathcal{G}$ be a family of recommender models. For any $\delta > 0$, with probability at least $1 - \delta$, Niu et al. (2016) demonstrate:*

$$
\sup_{g \in \mathcal{G}}|\hat{R}_{\mathrm{weak}}(g) - R(g)| \leq 4\kappa_{\mathrm{p}}L_{\ell}\mathfrak{R}_{\mathrm{p}} + 2L_{\ell}\mathfrak{R}_{\mathrm{u}} + 2\kappa_{\mathrm{p}}\sqrt{\frac{\ln(4/\delta)}{2n_{\mathrm{p}}}} + \sqrt{\frac{\ln(4/\delta)}{2n_{\mathrm{u}}}}.
$$

where $\mathfrak{R}_{\mathrm{p}}$ and $\mathfrak{R}_{\mathrm{u}}$ are the Rademacher complexities of $\mathcal{G}$ given $n_{\mathrm{p}}$ positive samples and $n_{\mathrm{u}}$ unlabeled samples, respectively. $L_\ell$ is the Lipschitz constant of $\ell$ with respect to $g$.

**Theorem 2** (Variance reduction). *The variance of $R_{\mathrm{weak}}$ is smaller than that of the ideal loss $R_{\mathrm{ideal}}$ given $\kappa_{\mathrm{p}} \leq 0.5$.*

*Proof.* The variance of the ideal loss can be expressed as

$$\mathrm{Var}(\hat{R}_{\mathrm{ideal}}(g)) = \left(\frac{\kappa_{\mathrm{p}}^2}{n_{\mathrm{p}}}\mathrm{Var}(\ell(g(x^{\mathrm{p}}), +1))\right) + \left(\frac{\kappa_{\mathrm{n}}^2}{n_{\mathrm{n}}}\mathrm{Var}(\ell(g(x^{\mathrm{n}}), -1))\right)$$

$$== \frac{\kappa_{\mathrm{p}}^2}{n_{\mathrm{p}}}\sigma_{\mathrm{p}}^2 + \frac{\kappa_{\mathrm{n}}^2}{n_{\mathrm{n}}}\sigma_{\mathrm{n}}^2,$$

where $\sigma_{\mathrm{p}}^2 = \mathrm{Var}(\ell(g(x^{\mathrm{p}}), +1))$, $\sigma_{\mathrm{n}}^2 = \mathrm{Var}(\ell(g(x^{\mathrm{n}}), -1))$. The variance of $\hat{R}_{\mathrm{weak}}(g)$ is:

$$\mathrm{Var}(\hat{R}_{\mathrm{weak}}(g)) = \left(\frac{\kappa_{\mathrm{p}}^2}{n_{\mathrm{p}}}\mathrm{Var}(\tilde{\ell}(g(x^{\mathrm{p}})))\right) + \left(\frac{1}{n_{\mathrm{u}}}\mathrm{Var}(\ell(g(x^{\mathrm{u}}), -1))\right)$$

$$= \left(\frac{\kappa_{\mathrm{p}}^2}{n_{\mathrm{p}}}\mathrm{Var}(\ell(g(x), +1)) + \frac{\kappa_{\mathrm{p}}^2}{n_{\mathrm{p}}}\mathrm{Var}(\ell(g(x), -1)))\right)$$

$$+ \left(\frac{1}{n_{\mathrm{u}}}\mathrm{Var}(\ell(g(x^{\mathrm{u}}), -1))\right)$$

$$= \frac{\kappa_{\mathrm{p}}^2}{n_{\mathrm{p}}}(\sigma_{\mathrm{p}}^2 + \sigma_{\mathrm{n}}^2) + \frac{1}{n_{\mathrm{u}}}\sigma_{\mathrm{n}}^2,$$

where the covariance between the two terms is zero, which immediately holds if positive and unlabeled samples are independently drawn.

To show that $\mathrm{Var}(\hat{R}_{\mathrm{weak}}(g)) \leq \mathrm{Var}(\hat{R}_{\mathrm{ideal}}(g))$, it is equivalent to show that

$$\frac{\kappa_{\mathrm{p}}^2}{n_{\mathrm{p}}}(\sigma_{\mathrm{p}}^2 + \sigma_{\mathrm{n}}^2) + \frac{1}{n_{\mathrm{u}}}\sigma_{\mathrm{n}}^2 \leq \frac{\kappa_{\mathrm{p}}^2}{n_{\mathrm{p}}}\sigma_{\mathrm{p}}^2 + \frac{\kappa_{\mathrm{n}}^2}{n_{\mathrm{n}}}\sigma_{\mathrm{n}}^2.$$

That is:

$$\frac{\kappa_{\mathrm{p}}^2}{n_{\mathrm{p}}} + \frac{1}{n_{\mathrm{u}}} \leq \frac{\kappa_{\mathrm{n}}^2}{n_{\mathrm{n}}}.$$

Since $n_{\mathrm{u}} >> n_{\mathrm{p}}, n_{\mathrm{n}}$, $1/n_{\mathrm{u}} \to 0$. Moreover, since $n_{\mathrm{p}}/n_{\mathrm{n}} = \kappa_{\mathrm{p}}/\kappa_{\mathrm{n}}$, the above inequality holds if:

$$\kappa_{\mathrm{p}} \leq \kappa_{\mathrm{n}},$$

which holds if $\kappa_{\mathrm{p}} \leq 0.5$, a condition naturally holds in most recommendation scenarios where the positive samples are sparse and much smaller than positive samples.

$\square$

**Theorem 3.** *Let $\mathcal{G}$ be a family of recommender models; $g^* = \arg\min_{g \in \mathcal{G}} R_{\mathrm{ideal}}(g)$ be the ground-truth optimal recommender acquired by minimizing the ideal risk (1), $g^\dagger = \arg\min_{g \in \mathcal{G}} \hat{R}_{\mathrm{weak}}(g)$ be the recommender acquired by minimizing (6). For any $\delta > 0$, with probability at least $1 - \delta$, we have*

$$R\left(g^\dagger\right) - R\left(g^*\right) \leq 8\kappa_{\mathrm{p}}L_\ell\mathfrak{R}_{\mathrm{p}} + 4L_\ell\mathfrak{R}_{\mathrm{u}} + \kappa_{\mathrm{p}}\sqrt{\frac{8\ln(4/\delta)}{n_{\mathrm{p}}}} + \sqrt{\frac{2\ln(4/\delta)}{n_{\mathrm{u}}}},$$

*where $\mathfrak{R}_{\mathrm{p}}$ and $\mathfrak{R}_{\mathrm{u}}$ are the Rademacher complexities of $\mathcal{G}$ given $n_{\mathrm{p}}$ positive samples and $n_{\mathrm{u}}$ unlabeled samples, respectively. $L_\ell$ is the Lipschitz constant of $\ell$ with respect to $g$.*

*Proof.* Based on Lemma 1, we have

$$
\begin{aligned}
R(g^\dagger) - R(g^*) &= \left( \widehat{R}_{\text{weak}}(g^\dagger) - \widehat{R}_{\text{weak}}(g^*) \right) \\
&\quad + \left( R(g^\dagger) - \widehat{R}_{\text{weak}}(g^\dagger) \right) + \left( \widehat{R}_{\text{weak}}(g^*) - R(g^*) \right) \\
&\leq 0 + \left( R(g^\dagger) - \widehat{R}_{\text{weak}}(g^\dagger) \right) + \left( \widehat{R}_{\text{weak}}(g^*) - R(g^*) \right) \\
&\leq 0 + 2\sup_{g \in \mathcal{G}} |\widehat{R}_{\text{weak}}(g) - R(g)| \\
&\leq 8\kappa_{\text{p}} L_\ell \mathfrak{R}_{\text{p}} + 4 L_\ell \mathfrak{R}_{\text{u}} + \kappa_{\text{p}} \sqrt{\frac{8\ln(4/\delta)}{n_{\text{p}}}} + \sqrt{\frac{2\ln(4/\delta)}{n_{\text{u}}}},
\end{aligned}
$$

$\square$

**Lemma 2.** *For any $\ell(\hat{y}, y)$ that is convex in $\hat{y}$, $\ell(\hat{y}, y) - \ell(\hat{y}, -y)$ is convex in $\hat{y}$ if and only $\ell(\hat{y}, y) - \ell(\hat{y}, -y)$ is linear in $\hat{y}$.*

*Proof.* The lemma comes from the Theorem 4.1 by Sugiyama et al. (2022). We provide a straight-forward proof here. Since $\ell(\hat{y}, y)$ is convex, we have:

$$
\ell(\hat{y_1} + \hat{y_2}, y) \leq \ell(\hat{y_1}, y) + \ell(\hat{y_2}, y)
$$

To make $\ell(\hat{y}, y) - \ell(\hat{y}, -y)$ convex, the term $-\ell(\hat{y}, -y)$ should be convex. Therefore, we have:

$$
\ell(\hat{y_1} + \hat{y_2}, y) \geq \ell(\hat{y_1}, y) + \ell(\hat{y_2}, y)
$$

Combining the two inequalities, we conclude that to make $\ell(\hat{y}, y) - \ell(\hat{y}, -y)$ convex, the sufficient and necessary condition is: $\ell(\hat{y_1} + \hat{y_2}, y) = \ell(\hat{y_1}, y) + \ell(\hat{y_2}, y)$, which indicates that $\ell(\hat{y}, y) - \ell(\hat{y}, -y)$ is linear in $\hat{y}$. $\square$

**Theorem 4.** *Consider a recommender $g$ where the output score $s$ is transformed by a sigmoid function such that: $g(x) = \text{sigmoid}(s(x))$. Consider the logistic error measure, the surrogate risk $R_{\text{weak}}(g)$ is convex with respect to the score $s$.*

*Proof.* The surrogate risk $R_{\text{weak}}(g)$ consists of two terms: $\hat{\ell}(g(x))$ and $\ell(g(x), +1)$. Under the logistic error measure, we have:

$$
\ell(g(x), +1) = -\ln(g(x)),
$$
$$
\ell(g(x), -1) = -\ln(1 - g(x)).
$$

Since $g(x) = \text{sigmoid}(s)$, the error measure can be rewritten as:

$$
\ell(s, y) = -\ln(\frac{1}{1 + e^{-ys}})
$$

First, we examine the convexity of the term $\ell(s, 1)$. The second derivative is:

$$
\frac{d^2\ell(s, +1)}{ds^2} = \frac{e^s}{(e^s + 1)^2} > 0 \tag{12}
$$

which means that $\ell(s, +1)$ is convex with respect to $s$.

Next, we analyze the term $\hat{\ell}(s)$, which can be expressed as:

$$
\begin{aligned}
\ell(s, +1) - \ell(s, -1) &= -\ln(\frac{1}{1 + e^{-s}}) + \ln(\frac{1}{1 + e^s}) \\
&= -\ln(\frac{1}{1 + e^{-s}}) + \ln(\frac{e^{-s}}{1 + e^{-s}}) \\
&= -s.
\end{aligned} \tag{13}
$$

Since $\hat{\ell}(s)$ is linear with respect to $s$, it is also convex according to Lemma 2. Finally, as the sum of two convex functions remains convex, $R_{\text{weak}}(g)$ is convex with respect to $s$.

Then, consider a matrix factorization model with $\mathbf{U}_x$ and $\mathbf{V}_x$ being the user embeddings and item embeddings indexed by $x$, respectively. The model is expressed as:

$$s(x) = \mathbf{U}_x^\top \mathbf{V}_x, g(x) = \text{sigmoid}(s(x)). \tag{14}$$

The score function $s$ is convex in model parameters ($\mathbf{U}_x$ and $\mathbf{V}_x$), which immediately follows from

$$\begin{aligned} \frac{d^2 s}{d\mathbf{U}_x^2} &= 0, \\ \frac{d^2 s}{d\mathbf{V}_x^2} &= 0. \end{aligned} \tag{15}$$

Therefore, $R_{\text{weak}}$ is convex in $\mathbf{U}_x$ and $\mathbf{V}_x$, which immediately follows from the composition of convex functions is convex, $R_{\text{weak}}$ is convex in $s$, and $s$ is convex in model parameters.

$\square$

**Lemma 3.** *Suppose $g$ is a recommender model based on matrix factorization, with the output score $s$ transformed by a sigmoid function, and the surrogate risk $R_{\text{weak}}(g)$ is convex with respect to the score $s$. The surrogate risk $R_{\text{weak}}(g)$ is minimized by the recommender $g$. Then, the minimization of the surrogate risk $R_{\text{weak}}(g)$ converges within a finite number of iterations.*

*Proof.* According to Theorem 4, the surrogate risk $R_{\text{weak}}(g)$ is convex with respect to the score $s$. Given the recommender $g$ as matrix factorization, $R_{\text{weak}}(g)$ is convex in the model parameters, which immediately follows from the composition of convex functions. Therefore, the minimization for the convex $R_{\text{weak}}(g)$ converges within a finite number of iterations. $\square$

### B.2. Solution to Proximal Transport Problem

The PT problem in (7) is a linear programming problem and can be solved with traditional linear programming solvers (Gurobi Optimization, 2020; Hart et al., 2011). Nevertheless, such solutions often overlook the unique structure of optimal transport problems, thereby suffering from relatively large cost. In this section, we demonstrate a reformulation that aligns PT with the well-established Kantorovich problem, which permits solution to PT problem using specialized solvers for the Kantorovich problem, such as iterative Bregman projections (Benamou et al., 2015), Sinkhorn iterations (Altschuler et al., 2017; Dvurechensky et al., 2018), and the network-simplex method (Flamary et al., 2021), with superior convergence rate and efficiency (Bonneel et al., 2011; Ling and Okada, 2007; Pele and Werman, 2009).

To reformulate the PT problem (7) as a Kantorovich problem, we introduce a slack variable $\mathbf{s} \in \mathbb{R}_+^m$, and rephrase the inequality constraint $\boldsymbol{\pi}^\top \mathbf{1}_n \leq w * \mathbf{b}$ in (7) as an equality constraint:

$$\boldsymbol{\pi}^\top \mathbf{1}_n \leq w * \mathbf{b} \Rightarrow \boldsymbol{\pi}^\top \mathbf{1}_n + \mathbf{s} = w * \mathbf{b}, \tag{16}$$

which yields the equality constraints of $\mathbf{b}$ as follow:

$$\begin{cases} \mathbf{1}_n^\top \boldsymbol{\pi} \mathbf{1}_m + \mathbf{s}^\top \mathbf{1}_m = (w * \mathbf{b})^\top \mathbf{1}_m \\ 1 + \mathbf{s}^\top \mathbf{1}_m = w * \|\mathbf{b}\|_1 \end{cases}. \tag{17}$$

Combining (17) and (7), we reformulate the PT problem (7) as:

$$\boldsymbol{\pi}^*(\alpha, \beta; w) := \arg\min_{\boldsymbol{\pi} \in \Pi(\alpha, \beta; w)} \langle \mathbf{C}, \boldsymbol{\pi} \rangle,$$

$$\Pi(\alpha, \beta; w) := \begin{cases} \boldsymbol{\pi} \in \mathbb{R}_+^{n \times m} : \boldsymbol{\pi} \mathbf{1}_m = \mathbf{a}, \mathbf{1}_n^\top \boldsymbol{\pi} \mathbf{1}_m = 1, \\ \mathbf{s} \in \mathbb{R}_+^m : \boldsymbol{\pi}^\top \mathbf{1}_n + \mathbf{s} = w * \mathbf{b}, \mathbf{s}^\top \mathbf{1}_m = w * \|\mathbf{b}\|_1 - 1 \end{cases}.$$

To substantiate the structural identity between the constraint set $\Pi(\alpha, \beta; w)$ in PT and that in the Kantorovich problem (Xu et al., 2020), we introduce augmented matrices and vectors as follows:

$$\tilde{\boldsymbol{\pi}} = \begin{bmatrix} \boldsymbol{\pi} & \mathbf{0}_{n \times 1} \\ \mathbf{s}^\top & \mathbf{0}_{1 \times 1} \end{bmatrix}, \quad \tilde{\mathbf{a}} = \begin{bmatrix} \mathbf{a} \\ w * \|\mathbf{b}\|_1 - 1 \end{bmatrix} \quad \text{and} \quad \tilde{\mathbf{b}} = \begin{bmatrix} w * \mathbf{b} \\ \mathbf{0}_{1 \times 1} \end{bmatrix}.$$

On the basis, we can reformulate $\Pi(\alpha, \beta; w)$ as

$$\tilde{\Pi}(\tilde{\alpha}, \tilde{\beta}; w) := \left\{ \tilde{\boldsymbol{\pi}} \in \mathbb{R}_+^{\tilde{n} \times \tilde{m}} : \tilde{\boldsymbol{\pi}} \mathbf{1}_{\tilde{m}} = \tilde{a}, \tilde{\boldsymbol{\pi}}^\top \mathbf{1}_{\tilde{n}} = \tilde{b} \right\} \tag{18}$$

which precisely mirrors the constraint set in the Kantorovich problem. Furthermore, we extend the transport cost matrix $\mathbf{C}$ to $\tilde{\mathbf{C}}$ to align with this formulation:

$$\tilde{\mathbf{C}} = \begin{bmatrix} \mathbf{C} & \xi \mathbf{1}_n \\ \xi \mathbf{1}_m & 2\xi + A \end{bmatrix}. \tag{19}$$

which incorporates a bounded scalar $\xi$ and a constant $A > \max(C_{ij})$. Xu et al. (2020) shows that the transport cost in the Kantorovich problem can be rewritten as $\left\langle \tilde{\mathbf{C}}, \tilde{\boldsymbol{\pi}} \right\rangle$. Therefore, the PT problem can be formulated as

$$\tilde{\boldsymbol{\pi}}^*(\tilde{\alpha}, \tilde{\beta}; w) = \arg\min_{\tilde{\boldsymbol{\pi}} \in \tilde{\Pi}(\tilde{\alpha}, \tilde{\beta}; w)} \left\langle \tilde{\mathbf{C}}, \tilde{\boldsymbol{\pi}} \right\rangle. \tag{20}$$

In summary, through these augmentations and reformulations, encapsulated in (20), we seamlessly recast the PT problem as a Kantorovich problem. This equivalence enables us to leverage existing optimal transport solvers for solving the PT problem, thereby offering computational advantages and methodological coherence.

