# OpenReview forum: "Unbiased Recommender Learning from Implicit Feedback via Weakly Supervised Learning"
_ICML.cc/2025/Conference — ICML 2025 poster_

### Official Review · Reviewer_Z2t6 · 2025-03-08

**Overall Recommendation:** 3

**Summary:**

This paper proposes a novel approach named positive-unlabeled recommender learning, to handling the challenge of missing negative feedback in implicit feedback recommendation systems. The key contribution is Progressive Proximal Transport (PPT), an optimal transport-based method that estimates the class prior by minimizing the transport cost between positive and unlabeled samples. The paper presents theoretical justifications for the proposed method and validates it through extensive experiments on three real-world datasets, demonstrating superior performance compared to existing methods.

**Claims And Evidence:**

Yes. PURL provides an unbiased and consistent risk estimator for implicit feedback recommendation. PPT effectively estimates the class prior without requiring heuristic assumptions or propensity scores. Empirical results show that PURL outperforms state-of-the-art baseline methods.

**Essential References Not Discussed:**

The references discussed some classical research directions of implicit feedback recommendation and MNF, but overlooked one direction: MNF in sequential recommendation. This is a very practical direction since people interact with recommender systems in a ordinal manner. Prior work like below discussed this direction which is not discussed in this paper.
- Wang, M., Gong, M., Zheng, X., & Zhang, K. (2018). Modeling dynamic missingness of implicit feedback for recommendation. Advances in neural information processing systems, 31.

**Experimental Designs Or Analyses:**

I've checked the experimental setting and their results & analysis.

**Methods And Evaluation Criteria:**

Yes, the chosen datasets are sufficient and the NDCG is a reasonable metric for recommendation tasks.

**Other Comments Or Suggestions:**

1. Could PPT be extended to handle multi-class implicit feedback scenarios instead of binary PU learning?
2. How sensitive is PURL’s performance to hyperparameter tuning, particularly in estimating the class prior?

**Other Strengths And Weaknesses:**

Strengths:
The use of optimal transport for class prior estimation is innovative. Unbiasedness and consistency of the estimator are formally proven, and the empirical results on real-world datasets are promising.

Weaknesses:
More discussion on scalability for large datasets is needed. In reality the data sparsity will reach 99% in recommender systems. How would proposed PPT perform is still unclear. The direct impact of PPT on performance is not fully isolated.

**Questions For Authors:**

n/a

**Relation To Broader Scientific Literature:**

The paper introduced progressive proximal transport to address the non-random missing problem, which is widely existing issue for broader research community. So the work has a potential of big impacts.

**Theoretical Claims:**

I've checked the derivation of the PURL risk estimator and its unbiasedness, finding the claims are sound.

---

> ### Author Rebuttal · Authors · 2025-04-01
>
> #### **[W1] More discussion on scalability.**
> **Response.** We express our sincere gratitude for this valuable comment. **We add a scalability analysis on the additional ML-1M dataset that is larger than the datasets involved in this paper.**
> In the below table, we summarize the performance of PURL given varying ratios of unlabeled samples (denoted as P).
>
> | P| 0.1 | 0.2 | 0.3 | 0.4 | 0.5 | 0.6 | 0.7 | 0.8 | 0.9 | 1 |
> |------|--|--|--|--|--|--|--|--|--|---|
> | NDCG\@1| 0.81572 | 0.81875 | 0.81841 | 0.82143 | 0.81320| 0.82093 | 0.81455 | 0.82244 | **0.82463** | 0.82059|
> | NDCG\@3| 0.80695 | 0.80828 | 0.81152 | 0.81185 | 0.80832 | 0.81150| 0.8101 0 | 0.81088 | 0.81037 | **0.81316**|
> | NDCG\@5| 0.80176 | 0.80649 | 0.80738 | 0.80680| 0.80483 | 0.80667 | 0.80835 | 0.80744 | 0.80868 | **0.80978**|
>
>
> As P increases, the performance of PURL generally increases with some fluctuations. The best performance is achieved when P>0.9, which is **consistent with the results reported in the main text and showcases scalability.** Compared to the results on smaller datasets, the performance is less sensitive to P. This can be attributed to the fact that even a small P in a large dataset corresponds to a large number of unlabeled samples, which suffices to meet the demand of unlabeled samples in Theorem 2 to reduce variance and improve overall performance.
>
> #### **[W2] The direct impact of PPT on performance.**
> **Response:** ! In this work, we introduce two components: (1) the PURL loss, which effectively leverages unlabeled samples; and (2) the PPT strategy, which accurately estimates class priors for calculating the PURL loss. To quantify the individual contributions of these components, we conducted a series of ablative experiments.
> - To discern the contribution of PURL, we varied the proportion of unlabeled samples (denoted as P) used in the PURL loss calculation, as illustrated in Figure 2. On both datasets,  the performance consistently improves as P increases, which demonstrates the effectiveness of PURL.
> - **To discern the contribution of PPT**, we compared the performance of PURL with PPT-estimated class prior and other values of class priors, in Table 3. The results show that the PPT-estimated class prior consistently led to the best recommendation performance. **This indicates that the PPT class prior is effective in estimating the class prior and thus improving the ImplicitRec performance.**
>
> #### **[Q1] Could PPT be extended to multi-class implicit feedback scenarios?**
> **Response.** We agree that **PPT can be extended to multi-class** scenarios. A plain approach would be treating the multi-value class labels as multiple binary classes. For example, if we have two positive classes (1, 2) and a negative class (0), we can treat them as two binary classes (0 v.s. 1 and 0 v.s. 2). The PPT can be applied to each binary class separately.
>
> #### **[Q2] Sensitivity to hyperparameter tuning.**
>
> **Response.**  Once again, we sincerely appreciate the reviewer's meticulous and constructive comments. We discuss the sensitivity to hyperparameter tuning in the following two aspects:
> - **Firstly, for the recommendation performance**, we conducted comprehensive experiments to investigate the sensitivity of key hyperparameters, including the ratio of unlabeled samples (Fig. 5), batch size (Fig. 7-a), learning rate (Fig. 7-b), and embedding dimension (Fig. 7-c). Different datasets exhibit varying sensitivity to these hyperparameters, which is largely dependent on the specific characteristics of the data. The detailed analysis is provided in Section 4.5 and Appendix B.3.
> - **Secondly, for the class prior estimation**, the PPT approach is generally isolated from these hyperparameters. To showcase this claim, we investigate the logs of sensitivity analysis that record the class prior estimation results $\hat{\kappa}_p$. Below are the results with different batch sizes, averaged over 5 random seeds:
>
> | Batch size | 256 | 512 | 1024 | 2048 |
> |--|--|--|--|--|
> | $\hat{\kappa}_p$| $0.0499_{\pm0.018}$ | $0.0541_{\pm0.0194}$ | $0.0607_{\pm0.0212}$ | $0.0499_{\pm0.0133}$ |
>
>  Across a wide range of batch sizes, the estimated class prior $\hat{\kappa}_p$  remains stable around 0.05 (i.e., the class prior producing the best recommendation performance in Table 3). It indicates that PPT is robust to variations in hyperparameters.
>
>
> #### **[References] MNF in sequential recommendation should be discussed.**
> **Response.** Thank you very much for your insightful comment. We agree that the MNF problem also matters in sequential recommendation. In the revised version, we will include discussions on key works in this area, including but not limited to [1,2,3].
>
> [1] Modeling dynamic missingness of implicit feedback for recommendation. NeurIPS. 2018.
>
> [2] Modeling dynamic missingness of implicit feedback for sequential recommendation. IEEE TKDE. 2020.
>
> [3] Sequential learning over implicit feedback for robust large-scale recommender systems. PKDD. 2019.

---

### Official Review · Reviewer_pJHQ · 2025-03-12

**Overall Recommendation:** 4

**Summary:**

- The study reframes implicit feedback recommendation as a weakly supervised learning problem, and introduces a model-agnostic framework, termed PURL, to handle the missing negative feedback problem.
- Central to this framework is the incorporation of the PU-learning method, which ensures unbiasedness given positive and unlabeled samples given accurate prior estimation.
- To achieve accurate prior estimation, authors proposed Progressive Proximal Transport (PPT), which estimates class priors by minimizing the proximal transport cost between positive and unlabeled data samples.
- To validate the efficacy of the proposed approach, extensive experiments were carried out across multiple datasets, yielding results that underscore its practical utility and effectiveness.

**Claims And Evidence:**

The proposed method provides a PU learning approach, which is feasible to handle this problem. Theoretical and empirical evidence is provided.

**Essential References Not Discussed:**

The section 'related works' discussed recent progress of PU learning and implicit feedback recommendation. Section 2.2 introduces optimal transport. Despite the primary references are covered, you can consider expanding the discussion to include broader applications of optimal transport in recommendation systems. This would provide a more comprehensive context for understanding the role and relevance of optimal transport within the field.

**Experimental Designs Or Analyses:**

The empirical investigation of this study is comprehensive, and the analysis makes sense.

**Methods And Evaluation Criteria:**

The proposed methods make sense and evaluation criteria follow the standard practice.

**Other Comments Or Suggestions:**

- Reference format should be checked. For example, you used "ACM Trans. Manage. Inf. Syst.", ISO abbreviation for journals. However, you used "NeurIPS" and "CIKM" for conferences, which are not ISO abbreviations.

- In section 4.4, you wrote "According to Section 4.4". Is there any reason to refer to the same section? Otherwise, it should be deleted.

- In line 75, you wrote "missing-not-at-random (MNF) issue". It should be "missing negative feedback (MNF) issue", right?

**Other Strengths And Weaknesses:**

Strengths
- This paper effectively highlights the significance of unlabeled data in recommendation systems.  An estimator that harness the unlabeled data for model training while ensuring unbiasedness is proposed.
- The theoretical properties of the proposed estimator is investigated. The method using OT for class prior estimation is fresh for me.
- According to Table 1-2, the proposed method exhibits superior empirical performance. Comprehensive empirical studies are conducted to validate the proposed method.

Weaknesses
- This paper has two main components. In this situation, an ablation study is critical to assess the contribution of each component to the overall performance. However, this study seems lacking.
- Some statements in this paper need further clarification. Please see the questions below.
- The originality of the proposed optimal transport problem--PPT--is not clarified. Authors should clarify the originality of PPT, or provide a concrete reference.
- The code link is provided, but the repository seems to be expired.

**Questions For Authors:**

- In line 40, you wrote "identifying accurate propensity scores remains an elusive goal in ImplicitRec". However, propensity score estimation is challenging in general, not only in implicit feedback recommendation. It does not interfere the success of propensity-based methods.
- In table 1-2, it seems that some methods using negative feedback are included. Ideally they should outperform implicit feedback methods, since the accurate negative feedback is available, is it right?
- I am curious about the connection between PURL and noisy label recommendation. Since the unlabeled samples are available, naively treating them as negative could lead to false labeling, which seems to be a noisy label problem. What is the advantage of formulating implicit feedback recommendation as a weak supervised learning problem over formulating it as anoisy label problem?

**Relation To Broader Scientific Literature:**

In terms of methodology, this paper is related to the field of PU learning and optimal transport. In terms of application, it is related to the field of recommendation systems with implicit feedback.

**Theoretical Claims:**

Theories are clear to follow and the sources from PU learning literature are cited.

---

> ### Author Rebuttal · Authors · 2025-04-01
>
> #### **[W1] Ablation study matters.**
> **Response:** Thank you for your kind comment! In this work, we devise two components: (1) the PURL loss; (2) the PPT strategy. **We have conducted experiments to discern their individual contributions**.
> - To discern the contribution of PURL that leverages unlabeled samples, we have conducted ablative experiments by changing the proportion of unlabeled samples involved in Fig.2. The results show that on both datasets, the performance increases with the proportion of involved unlabeled samples, which demonstrates the effectiveness of PURL in leveraging unlabeled samples.
> - To discern the contribution of PPT that estimates class priors, we have conducted ablative experiments in Table 3. The results show that the class prior estimate of PPT, with the smallest $\mathbb{W}$, corresponds to the best recommendation performance.
>
>
> #### **[W2, Q1] Challenge of propensity estimation.**
> **Response:** Thank you for your insightful comment. We agree that the propensity score estimation is a general challenge, but we demonstrate that this challenge is exacerbated in ImplicitRec, for the following reasons:
>
> - In recommendation, the propensity score is typically the probability of a specific treatment given a user-item pair.
> - In ImplicitRec, this probability is hard to estimate due to the absence of negative feedback. For example, in CTR prediction, the propensity score is the exposure probability of a user-item pair. However, ImplicitRec only provides positive (exposure & click) and unlabeled samples (non-exposure), lacking negative samples (exposure & non-click). This makes estimating exposure probability infeasible.
> - This challenge is unique to ImplicitRec and significantly hampers the effectiveness of existing propensity-based methods in achieving unbiased recommendations in ImplicitRec.
>
> #### **[W2, Q2] Why does ImplicitRec model outperform some explicit models?**
> **Response:** Thank you for your insightful comments. The phenomenon is noticable and we justify it from two aspects.
> - **Theoretical aspect.** The proposed PURL loss $R_\mathrm{purl}$ **reduces the variance** of the ideal binary loss $R_\mathrm{ideal}$ that uses explicit feedback, **given $\kappa_p\leq 0.5$ (a naturally holding condition) and large number of unlabeled samples.** The variance reduction property is justified in Theorem 2, which leads to better generalization and performance.
> - **Empirical aspect.** Figure 5 shows consistent performance improvement with more unlabeled samples, validating the theoretical aspect above. Thus, more unlabeled samples decrease PURL variance and improve performance over explicit feedback methods.
>
> #### **[W2, Q3] The connection with noisy label recommendation.**
> **Response.** Yes, it is feasible to employ learning-from-noisy-label techniques to address the MNF issue within ImplicitRec. However, the weakly supervised learning framework used in this work is more suitable for two main reasons:
> - **Theoretical supports**. Most noisy label methods are heuristic or rely on strong assumptions about the noise mechanisms. In contrast, the weakly supervised learning framework provides theoretical guarantees without such harsh assumptions.
> - **Empirical supports**. One exemplar work approaching ImplcitRec as learning from noisy labels is T-CE [1]. Our experiments below show that PURL outperforms T-CE, demonstrating the effectiveness of using weakly supervised learning for ImplicitRec.
>
> | Dataset | Model  | NDCG@1 | NDCG@3 | NDCG@5 | Recall@1 | Recall@3 | Recall@5 |
> |--|--|--|--|---|---|--|----|
> | Yahoo   | RecVAE | 0.65   | 0.683  | 0.719  | 0.054 | 0.149 | 0.234 |
> |  | T-CE   | 0.749  | 0.785  | 0.817  | 0.125 | 0.252 | 0.333 |
> |  | PURL   | 0.784  | 0.814  | 0.843  | 0.146 | 0.28 | 0.351 |
> | Coat    | RecVAE | 0.465  | 0.499  | 0.545 | 0.096 | 0.267 | 0.389    |
> |  | T-CE   | 0.448  | 0.47   | 0.509  | 0.081 | 0.228 | 0.347 |
> | | PURL   | 0.552  | 0.555  | 0.589  | 0.131 | 0.302 | 0.425 |
> | Kuairec | RecVAE | 0.422  | 0.434  | 0.448 | 0.11 | 0.183 | 0.214 |
> |  | T-CE   | 0.418  | 0.429  | 0.435  | 0.108 | 0.214    | 0.264    |
> |   | PURL   | 0.498  | 0.486  | 0.488  | 0.137  | 0.245    | 0.296    |
>
> #### **[W3] Originality of PPT.**
> **Response:** Thank you for the reminder. The PPT generalizes the canonical OT problem by relaxing the equality constraint for $\beta$ and changes its total mass from 1 to $w$. This formulation is specifically designed for ImplicitRec and, to our knowledge, has not been previously studied.
>
> #### **[W4] Project website.**
> **Response:** Thank you for your kind reminder. The repository has now been restored. Additionally, we offer a Docker file to facilitate quick reproduction.
>
>
> #### **[Reference and Other comments].**
> **Response.** Thank you for your meticulous comment. We will fix the typos, remove the invalid reference, adjust the related works and unify the reference formats in revision.
>
> [1] Denoising implicit feedback for recommendation. WSDM. 2021.

---

> > ### Comment · Reviewer_pJHQ · 2025-04-02
> >
> > Thank you for your additional experiments. The informative response has addressed my questions and I have accordingly adjusted my scores. Please add these discussions in revision.

---

### Official Review · Reviewer_t2jt · 2025-03-13

**Overall Recommendation:** 3

**Summary:**

The paper addresses the challenge of missing negative feedback (MNF) in implicit feedback-based recommender systems. Traditional approaches often rely on negative sampling, which risks misclassifying positive samples as negative, leading to bias and performance degradation. The authors propose PURL (Positive-Unlabeled Recommender Learning), a framework that treats implicit recommendation as a weakly supervised learning task, thereby eliminating the need for negative samples.

**Claims And Evidence:**

- PURL estimator is unbiased relative to the ideal risk
evidence: C.1 Theorem 1.
- Progressive Proximal Transport (PPT) accurately estimates class priors
evidence: Empirical results
-  PURL outperforms existing implicit feedback methods

**Essential References Not Discussed:**

I'm not familiar enough with the literature, so I don't see essential reference not discussed.

**Experimental Designs Or Analyses:**

I think the experiment is comprehensive: 13 baselines, covering pointwise, pairwise, and unbiased methods, which are commonly used in implicit feedback recommendation research.
The dataset is also diverse: Yahoo! R3 and Coat are explicit feedback datasets converted to implicit feedback KuaiRec is a fully implicit feedback dataset.

**Methods And Evaluation Criteria:**

The paper introduces PURL (Positive-Unlabeled Recommender Learning) and Progressive Proximal Transport (PPT) as core components to address missing negative feedback (MNF) in implicit recommendation. I think the proposed method is theoretically and empirically sound.

The evaluation metric include NDCG@1 NDCG@3 NDCG@5, I think make sense. recall is also measured.

**Other Comments Or Suggestions:**

See Above

**Other Strengths And Weaknesses:**

Overall, I think this is a good paper and all the claim are solid and sound for me. But I'm not working on the field for a while, so please also refer to other reviwer's suggestions.

**Questions For Authors:**

A small question, will using Yahoo! R3 and Coat that transfer rating to implicit feedback influence the evaluation?

**Relation To Broader Scientific Literature:**

I believe the proposed method contributes to the literature as a model-agnostic framework that reframes implicit feedback recommendation as a weakly supervised learning task, eliminating the need for negative samples.

**Theoretical Claims:**

I think the theretical claims are well proved. I'm not an expert for theory, therefore please also refer to other reviewer's suggesitons.

---

> ### Author Rebuttal · Authors · 2025-04-01
>
> **Thank you very much for your kind and sincere words. It is impressive to meet a review that candidly express limitations while responsibly and analyzing the claims and details of the paper. We truly appreciate it!**
>
>
> We have noticed the question about the dataset generation process and are happy to provide a detailed response.
> - Firstly, we chose the Yahoo and Coat datasets, which are commonly used in prior ImplicitRec studies such as CUMF [1], UBPR [2], and UPL [3].
> - In the training set, we retained only user-item pairs with positive feedback and treated those with negative feedback as unlabeled. This reflects real-world implicit feedback scenarios where only positive interactions are observed, and the rest are unlabeled. In the test set, we preserve the explicit feedback which is necessary to calculate the metrics (NDCG, Recall, etc.), mirroring the online serving scenario of implicit feedback recommendation.
> - This process aligns with common practices in ImplicitRec research [1-3].
>
> **Reference.**
>
> [1] Yuta Saito, Suguru Yaginuma, Yuta Nishino, Hayato Sakata, and Kazuhide Nakata. 2020. Unbiased Recommender Learning from Missing-Not-At-Random Implicit Feed- back. In WSDM. ACM, 501–509.
>
> [2] Yuta Saito. 2020. Unbiased Pairwise Learning from Biased Implicit Feedback. In ICTIR. ACM, 5–12.
>
> [3] Yi Ren, Hongyan Tang, Jiangpeng Rong, and Siwen Zhu. 2023. Unbiased Pairwise Learning from Implicit Feed- back for Recommender Systems without Biased Variance Control. In SIGIR. ACM, 2461–2465.

---

### Official Review · Reviewer_bC89 · 2025-03-13

**Overall Recommendation:** 4

**Summary:**

This paper focuses on addressing the lack of negative feedback in recommendations with implicit feedback and points out the limitations of recent unbiased estimator-based methods in identifying propensity scores and non-negative estimates. It proposes a novel positive-unlabeled recommender learning (PURL) framework, in which the core idea is to introduce a new estimator and use progressive proximal transport (PPT) to estimate the required class priors of unobserved samples. Extensive experiments are conducted on three public datasets to verify the effectiveness of PURL.

**Claims And Evidence:**

The claims made in the submission are supported by clear and convincing evidence; the claimed problem (MNF) is a common problem in recommendation systems with implicit feedback.

**Essential References Not Discussed:**

The literature discussion in the main text is a little brief.
It is better to move the discussion of related work from the appendix to the main text.

**Experimental Designs Or Analyses:**

I have checked the soundness/validity of any experimental designs or analyses. The experiments validate the overall performance of the proposed method.

**Methods And Evaluation Criteria:**

The proposed methods and/or evaluation criteria (e.g., benchmark datasets) make sense.

**Other Comments Or Suggestions:**

The consistency of the terms used needs improvement. For example, Are the user-item pair, user-item intersection, and user-item interaction the same thing?

**Other Strengths And Weaknesses:**

Strengths:
1. This paper targets missing negative feedback, an important issue in the field of implicit feedback recommendation. It also widely exists in real-world applications.
2. The proposed method is promising to tackle the issue. It contains a creative fusing of OT and weakly supervised learning and an important application in recommendation.
3. Diagrams are well prepared to facilitate understanding of important concepts and showcase efficacy.

Weaknesses:
1. The mass weight's connection to class prior estimation is not explicit, but it is a critical aspect to understand the subsequent contents.
2. The claim that Bayesian Personalized Ranking (BPR) lacks a solid theoretical foundation requires elaboration.
3. This paper utilized PPT, an OT-based approach for class prior estimation. However, OT is often costly in terms of both computation time and memory, as well as poor scalability w.r.t. the number of decision variables, which is quadratic to the batch size. Regrettably, in real-world recommendations, large batch size is often used in training recommendation systems for daily updates.  In this case, PPT might require longer training time and larger memory requirement.
4. The principle of dataset selection should be explained. In this paper, the selected datasets are Yahoo, Coat, and Kuairec. Why not select some standard datasets such as ML-10M?
5. Some steps in the derivation are difficult to follow. e.g., the derivation in equation 7. Necessary conditions and explanations are expected to facilitate readers to follow and check the derivation.

**Questions For Authors:**

See the above two parts.

**Relation To Broader Scientific Literature:**

This paper focuses on the problem of missing negative feedback, an issue seen in the literature of implicit feedback recommendation. The proposed PURL is connected to the method from the weakly supervised learning field. The proposed PPT is an optimal transport approach with modification for achieving class prior estimation.

**Theoretical Claims:**

I have checked the correctness of any proofs for theoretical claims. Some obstacles are presented in the questions below.

---

> ### Author Rebuttal · Authors · 2025-04-01
>
> **We sincerely appreciate the reviewer’s great efforts and insightful comments to improve our manuscript. In below, we address the raised concerns point by point and try our best to clarify any confusions.**
>
> #### **[W1] The connection between mass weight $w$ to class prior estimation is not explicit.**
> **Response.**  We agree that the connection needs clarification.
> - **Connection clarification.** We estimate the class prior as the inverse of the mass weight, i.e., $\hat{\kappa}_p=w$.
> - **Rationale of the connection.** The PT problem is designed to match the positive samples ($\alpha$) with a selective set including $1/w$ of the unlabeled samples ($\beta$). If there exist negative samples in the selective set, the PT discrepancy will be large. By finding the $w^*$ with minimum discrepancy, we can estimate the class prior $\hat{\kappa}_p = 1/w^*$. This process is visualized in Fig. 1.
> - **Toy example.** We provide simulated results in Fig.4. The ground-truth class prior $\kappa=0.5$. When we set $w=2$, all positive samples are exactly matched with the proximal positive samples within the unlabeled population, producing the minimal transport cost.
> #### **[W2] The claim concerning BPR requires elaboration.**
> **Response.** Thank you for your insightful comment. We elaborate the claim as follows, and promise to add it in revision.
> - The BPR loss randomly samples unlabeled samples as negatives. However, **not all unlabeled samples are true negatives.** Some may simply be items the user hasn't encountered yet. This misclassification makes BPR biased.
> - In contrast, **our proposed PURL loss is unbiased** given accurate class prior estimation. This is supported by Theorem 1 in our paper, which provides a theoretical support for the advantage of the proposed PURL.
>
> #### **[W3] Computational time and memory requirement.**
> **Response.** Thank you for your insightful question. We add the analysis on computational time and memory consumption below.
>
> - The complexity of solving the proposed PPT problem is mainly determined by the number of samples. We conducted an empirical study on its running time, revealing that, although the theoretical complexity remains not straightforward to derive, the **running time scales linearly with the number of samples on a log-log scale.** In the **largest test with batch size of 2,048**—exceeding typical batch sizes in most practices—the running time was **under 0.1 seconds**, rendering it negligible compared to model training.
> - The memory consumption is also affected by the number of samples. **We add experiments where we used the decorator profile in the memory_profiler package to test the memory consumption** in solving the PT problem. According to the results below, **in the largest setting of 2048 samples, the memory consumption is 52.9 MB**, which is acceptable and even negligible compared to the memory cost of training typical recommendation system.
>
> | Batch size | 128 | 256 | 512 | 1024 | 2048 |
> |------------|-----|-----|-----|------|------|
> | Memory consumption | 1.2Mb | 4Mb | 10.5Mb | 20.7Mb | 52.9Mb |
>
> - We promise to adding these discussions in the revised manuscript.
>
> #### **[W4] The principle of dataset selection should be explained.**
> **Response.** Thank you for your kind reminder. We select the datasets based on the following principles:
> - Firstly, we opt for the datasets that contain unbiased test set with explicit feedback. This is crucial for evaluating the performance of our method, where the explicit feedback is used to calculate the metrics (NDCG, Recall, etc.), and the unbiased set mirrors the online serving scenario.
> - Secondly, we follow the practice of previous works in the ImplicitRec field. Specifically, the employed Yahoo and Coat datasets were used in CUMF (WSDM'20), UBPR (SIGIR'20), UPL (SIGIR'23), CDR (ICLR'23) and SDR (ICLR'23).
> - Moreover, motivated by UIDR (ICLR'24), we add an additional dataset--Kuairec--to provide comprehensive evaluation of the proposed method on large-scale industrial dataset.
>
> #### **[W5] The derivation of Eq.7.**
> **Response.** Eq. 7 defines a proximal transport problem, which is a generalization of the canonical OT problem in Eq. 2.
> - To derive Eq. 7, we replace the marginal constraint $\boldsymbol{\pi}^\top \boldsymbol{1}_n=\mathbf{b}$ with $\boldsymbol{\pi}^\top \boldsymbol{1}_n \leq w \mathbf{b}$ with $w\geq 1$, which acquires a semi-relaxed OT.
> - On the basis, we add the constraint $ \boldsymbol{1}_m\boldsymbol{\pi}^\top \boldsymbol{1}_n = 1$, which forces the full mass to be transported to be 1.
> - Then, given the assumption that the mass in $\alpha$ and $\beta$ is uniformly distributed and sums to 1, we have $\mathbf{a}=\boldsymbol{1}_n$ and $\mathbf{b}=\boldsymbol{1}_m$. The derivation of Eq. 7 is completed.
>
> #### **[Other comments] Consistency of terms.**
> **Response.** Once again, thank you for your valuable comment. They refer to the same thing and we will use a unified term "user-item pair" in revision.

---

> > ### Comment · Reviewer_bC89 · 2025-04-05
> >
> > Thanks for the author's detailed reply, which addresses my concerns and questions. Accordingly, I am happy to vote for acceptance and slightly raise the score.

---

### Decision · Program_Chairs · 2025-05-01

**Decision:**

Accept (poster)

**Comment:**

This paper presents positive-unlabeled recommender learning (PURL), aiming to address the ambiguity of the missing entries (0's) in implicit feedback data (a 0 can mean either negative feedback or not aware of the item). Overall the paper received positive scores. All the reviewers acknowledge the importance of the studied problem and the proposed solution appears theoretically sound, and supported by empirical evidence. There were some shared concerns regarding the scalability of the optimal transport component of the method, which are largely resolved during the author-reviewer discussion phase.

However, after reading all the reviews, I am not sure if some of the reviewers' expertise perfectly aligns with this paper. Therefore, I read the paper very carefully myself (plus getting some external opinions from people whose research is on this very topic).

Let me start with the strengths: this paper tackles the unbiased recommender systems learning problem from an alternative angle as a weakly supervised learning problem (prior work often frames the task from a causal inference perspective and replies on some kind of propensity adjustment which can often be problematic), and consequently introduces an objective function that doesn't involve any propensity weighting, instead depends on the positive class prior $p(y=+1)$. The paper then revolves around estimating this positive class prior using an optimal-transport based approach. The empirical results demonstrate that the proposed method outperforms IPS-based baselines on three standard benchmark datasets for this kind of setting (Yahoo music, coat, and KuaiRec).

Regarding the weaknesses: I think the biggest one would be the assumption made in Section 3.1 where the unlabeled data distribution $p_u(x)$ follows the same distribution as the underlying data distribution $p_{data}(x)$. I am not convinced that this assumption is entirely correct. Intuitively this implies the unlabeled data has the same ratio of positive and negative as the (idealized) fully-observed data. However, we know the negative remains unchanged between the unlabeled data and the fully-observed data, but positive from the fully-observed data gets "distorted" by us observing some of the positive interactions. So in some sense, the more positive interactions we observe, the less correct this assumptions will be -- consider the extreme case of observing all the positive, then unlabeled data would consist of only negative which clearly does not follow the same distribution as $p_{data}(x)$. Fortunately typical RecSys dataset is highly-sparse hence this assumption is probably approximately correct. However, I wonder if there is any existing work validating this assumption, and if not, I feel that more emphasis should be put on addressing this.

Another weakness as pointed out by many other reviewers is the scalability. The authors did provide some evidence that the algorithm is capable of running on small-sized datasets with simple matrix factorization model. How would it scale to large-scale datasets/more complex models which are commonplace these days for recommender systems research?

Finally, a comment regarding the experimentation setup: it seems that all the algorithms are optimized with SGD. However, I think it has been observed that if a closed-form alternating least-square-style solution is available, it can often outperform SGD by a rather big margin. And I believe for some of the baselines, it is possible to run ALS-style optimization. How would the proposed method compare in that case?

In summary, I believe this paper certainly has enough strengths but I don't think it is as solid as its current scores indicate. Therefore, I am voting for a weak accept and hope the authors can take the feedback into account.

- One minor comment: I noticed in the experimental section, WMF (Hu et al. 2008) is listed as "unbiased" point-wise method. I don't think that is correct since it downweights 0's uniformly.